# A versatile regulatory toolkit of arabinose-inducible artificial transcription factors for Enterobacteriaceae

Gita Naseri [1,2✉], Hannah Raasch [2], Emmanuelle Charpentier[1,2] & Marc Erhardt [1,2✉]

The Gram-negative bacteria *Salmonella enterica* and *Escherichia coli* are important model organisms, powerful prokaryotic expression platforms for biotechnological applications, and pathogenic strains constitute major public health threats. To facilitate new approaches for research and biotechnological applications, we here develop a set of arabinose-inducible artificial transcription factors (ATFs) using CRISPR/dCas9 and *Arabidopsis*-derived DNA-binding proteins to control gene expression in *E. coli* and *Salmonella* over a wide inducer concentration range. The transcriptional output of the different ATFs, in particular when expressed in *Salmonella* rewired for arabinose catabolism, varies over a wide spectrum (up to 35-fold gene activation). As a proof-of-concept, we use the developed ATFs to engineer a *Salmonella* two-input biosensor strain, SALSOR 0.2 (SALmonella biosenSOR 0.2), which detects and quantifies alkaloid drugs through a measurable fluorescent output. Moreover, we use plant-derived ATFs to regulate β-carotene biosynthesis in *E. coli*, resulting in ~2.1-fold higher β-carotene production compared to expression of the biosynthesis pathway using a strong constitutive promoter.

[1] Max Planck Unit for the Science of Pathogens, Charitéplatz 1, 10117 Berlin, Germany. [2] Institut für Biologie, Humboldt-Universität zu Berlin, Philippstrasse 13, 10115 Berlin, Germany. ✉email: naseri@mpusp.mpg.de; marc.erhardt@hu-berlin.de

The Gram-negative bacteria *Salmonella enterica* and *Escherichia coli* are promising expression platforms for high-titer production and secretion of difficult-to-express proteins[1–4]. In addition, pathogenic strains of *S. enterica* and *E. coli* are among the world's most significant public health problems[5]. However, the complex mechanisms of the underlying gene regulatory networks (GRNs) controlling expression of many important virulence factors are poorly understood[6–8]. To understand such regulatory networks[9], and implement bacterial cells for biotechnological applications, robust genetic tools such as orthogonal transcriptional regulators that allow to artificially control gene expression are highly desirable[10]. Toward this goal, synthetic promoters such as BglBricks promoters[11] and "pro" series (*PproA*, *B*, *C*, and *D*)[12] were developed to modulate the expression of endogenous or heterologous proteins in *E. coli*. Later, Cooper et al. applied the "pro" series of constitutive promoters to improve the tunability of protein expression in *Salmonella*[13]. However, employing constitutive elements to modulate the gene expression may impose a metabolic burden on the cell because the expression of heterologous proteins competes with other cellular processes and may be undesirable when an orthogonal (minimal interference with the native cellular processes) and controllable (allowing expression at the desired time) system is required[14]. To address the aforementioned challenges, artificial transcription factors (ATFs), allowing for temporal, tight, and tunable gene expression, are favorable. While several ATFs have been developed for synthetic biology applications in eukaryotic organisms[15,16], only a limited number of ATFs are available for Gram-negative bacteria[17–20]. Ho et al. developed several CRISPR/dCas9-derived ATFs for synthetic biology applications in *E. coli*, from which one was adopted for activation of gene expression in *Salmonella*, resulting in a 3.7-fold induction[18]. However, to our knowledge, no comprehensive collection of inducible ATFs to control gene expression in *Salmonella* has been reported. In addition, few inducible ATFs, based on catalytically inactive CRISPR-associated protein Cas9 (dCas9) and Cas12a (dCas12) DNA-binding domains (DBDs) have been developed, e.g., in *E. coli*[17–20] and *Paenibacillus polymyxa*[21]. These ATFs are equipped with activation domains (ADs), e.g., the omega subunit of RNA polymerase (RNAP)[19] or bacterial enhancer-binding proteins (bEBPs)[20] to activate the bacterial endogenous transcriptional machinery. However, these systems require specific genetic backgrounds that harbor deletions of the omega subunit or bEBPs, which may lead to fitness

defects and might be unfavorable as the required genomic manipulation is restricted to specific host backgrounds. To overcome these issues, Dong et al. developed ATFs equipped with the AD of the bacterial transcription factor SoxS to activate gene expression in *E. coli*[17].

Here, we adopted and expanded the approach established by Dong et al., employing dCas9 DBD and SoxS AD to create tunable ATFs for gene expression modulation in *Salmonella*. Additionally, we explored a new class of ATFs, featuring heterologous DBDs derived from plant-specific transcription factors (TFs) of *Arabidopsis thaliana*[15], combined with the SoxS AD[17], to broaden our ATF library for inducible gene expression in Gram-negative bacteria. Expression of ATFs is typically controlled by exogenous inducers. While extensive research has been conducted on the arabinose-inducible *araBAD* promoter ($P_{BAD}$) in *E. coli*[22,23], less focus has been directed towards the $P_{BAD}$ expression system in *Salmonella*. Due to the slight differences in the arabinose induction system between the two species[24,25], we first genetically engineered the L-arabinose (hereafter, arabinose) catabolic pathway in *Salmonella enterica*, resulting in ~4.5- to ~6.5-fold enhancement in $P_{BAD}$-driven gene expression. We then determined an optimal 'arabinose induction window' (0.01% to 0.1% arabinose) for our ATF collection in *Salmonella*, achieving high heterologous gene expression while minimally impacting bacterial growth. As a proof-of-concept, we employed the developed ATFs to engineer a *Salmonella* strain as a sensitive biosensor for alkaloid drugs and an *E. coli* strain as a microbial cell factory for β-carotene production. The arabinose-inducible ATFs, presented in this study, augment the synthetic biology toolkit of bacterial transcriptional regulatory modules, enabling fine-tuned protein expression in the Gram-negative model organisms *Salmonella* and *E. coli*.

## Results

**Design of arabinose-inducible artificial transcription factors**. A core objective of our work was to develop inducible, heterologous regulators capable of genetically reprogramming gene regulatory networks in Enterobacteriaceae, specifically *Salmonella* and *E. coli*. To achieve this, we designed arabinose-inducible ATFs incorporating diverse DBDs originating from CRISPR/dCas9 and plant heterologous TFs (Fig. 1). In order to evaluate the performances of these ATFs to induce gene expression, we developed a set of reporter and expression plasmids. Chromosomally or

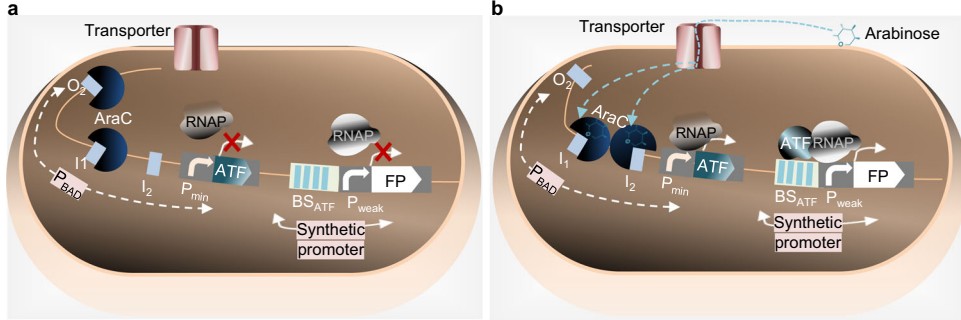

**Fig. 1 Principle of arabinose-inducible ATFs established in this study. a** Inducer-OFF state: In the absence of arabinose, an AraC dimer binds to the $O_2$ and $I_1$ half-sites, causing DNA looping, which prevents RNAP from accessing the promoter. As a result, the ATF is not expressed. Therefore, the expression of reporter FP that is controlled by the ATF is not induced. **b** Inducer-ON state: Externally added arabinose enters the cell via a transporter. The arabinose-bound AraC dimer changes conformation and binds the $I_1$ and $I_2$ half-sites of $P_{BAD}$, activating the transcription of the ATF. The ATF targets its BS(s) within a synthetic promoter, controlling FP expression. The interaction of the ATF and RNAP leads to increased FP expression from the synthetic promoter compared to the OFF state[26]. To simplify the figure, the RBSs and terminators located, respectively, upstream and downstream of the ATF and FP are not shown. ATF artificial transcription factor; BS binding site; FP fluorescent protein; $I_1$, $I_2$, and $O_2$ represent DNA binding half-sites.; $P_{BAD}$, arabinose-inducible *araBAD* promoter; $P_{min}$, minimal synthetic promoter containing the −35 and −10 essential elements; $P_{weak}$, weak promoter; RNAP RNA polymerase.

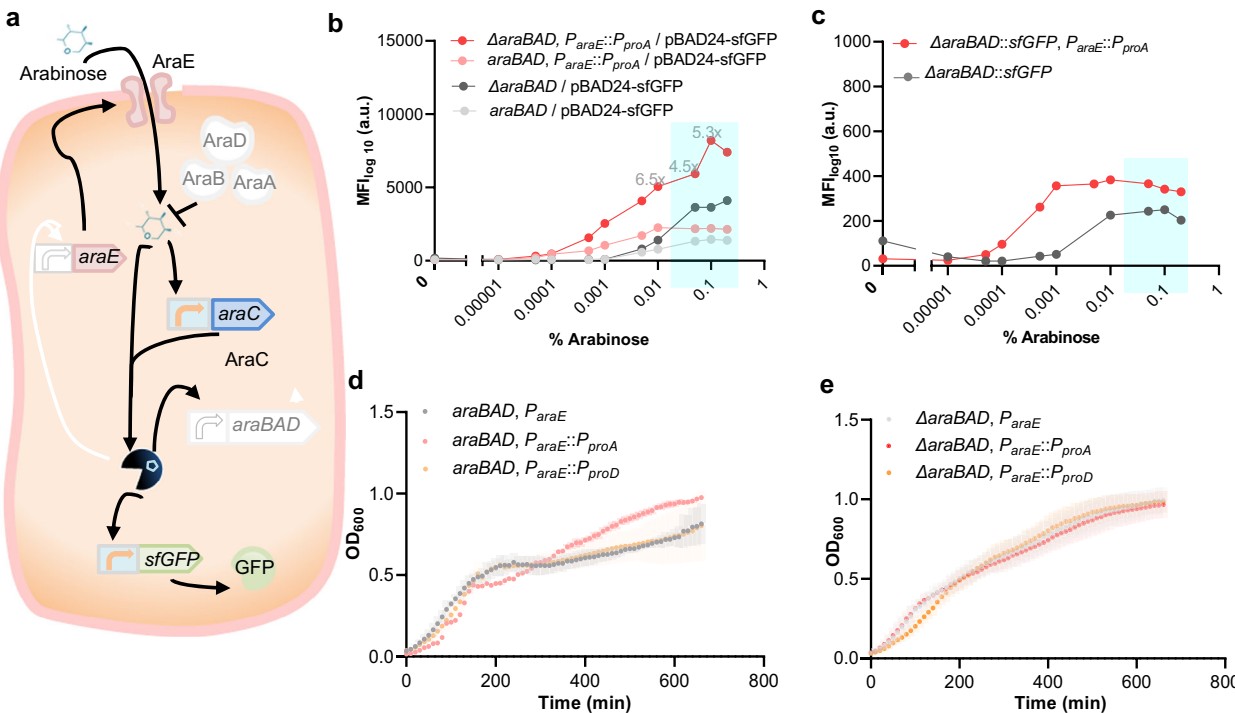

**Fig. 2 Optimization of the arabinose-based expression system in *Salmonella*. a** Regulatory network of the native arabinose utilization system in *S. enterica*[24], including the reporter cassette used in this study. The system consists of *araE*, encoding the arabinose transporter AraE essential for arabinose uptake, the *araBAD* operon, encoding enzymes for arabinose catabolism, and *araC*, encoding the regulator AraC. In the presence of high intracellular amounts of arabinose, AraC stimulates expression from the promoters $P_{BAD}$ and $P_{araE}$ (in the absence of arabinose, AraC represses $P_{BAD}$-derived expression), as well as its own expression. As a reporter, sfGFP, under the control of the $P_{BAD}$ promoter, was used. Indicated in white shading are the mutants characterized in this study: deletion of the *araBAD* operon and constitutive *araE* expression in different levels. To simplify the figure, the possible negative feedback of AraC is not shown, as it mainly contributes to providing a constant AraC level[51]. Characterization of dose-dependency of arabinose-induced gene expression from a plasmid (**b**) or chromosome (**c**) in *Salmonella*. Light grey, wild-type; dark grey, *araBAD* deletion mutant; light red, mutant with decoupled arabinose-dependent transporter–reporter system; dark red, mutant with *araBAD* deletion and decoupled arabinose-dependent transporter-reporter system. sfGFP-expression was measured by flow cytometry in the presence of 0.00001%, 0.00005%, 0.0001%, 0.0005%, 0.001%, 0.005%, 0.01%, 0.05%, 0.1%, and 0.2% arabinose added to the LB medium. The 'induction window' is highlighted in blue. "*x*" on top of the columns represents the fold induction compared to wild-type *S. enterica*. Each data point represents the mean of fluorescence intensity for 10,000 cells per sample. Growth curves of strains mutated for arabinose-independent expression of the transporter (**d**) and arabinose-independent expression of the transporter and deleted arabinose catabolism system (**e**) in the presence of 0.2% arabinose in *S. enterica* background. The growth of cells expressing *araE* from the weak $P_{proA}$ (light red)[13] or strong $P_{proD}$ (light orange)[13] constitutive promoter was compared to that of cells with arabinose-dependent *araE* expression from the native $P_{araE}$ in cells with or without deleted *araBAD* operon. The data shown are the average of three biological replicates. a.u. arbitrary units; MFI mean fluorescence intensity; $P_{BAD}$, arabinose-responsive promoter; OD$_{600}$, optical density at 600 nm; sfGFP superfolder green fluorescent protein. The full data are shown in Supplementary Data 1.

plasmid-encoded ATFs were placed under the control of $P_{BAD}$. In the absence of arabinose, dimeric AraC acts as a repressor of $P_{BAD}$, where one monomer binds to the operator $O_2$, and another monomer binds to the $I_1$ half-site in $P_{BAD}$, which results in the formation of a DNA loop that prevents RNAP from binding to $P_{BAD}$. In the presence of arabinose, and upon binding of arabinose to AraC, binding of the AraC-arabinose complex to the $I_2$ half-site in $P_{BAD}$ is allosterically induced, while binding to $O_2$ is decreased. In this configuration, AraC acts as an activator, promoting the binding of RNAP to $P_{BAD}$, which activates the expression of the ATF. Next, the ATF targets its binding site(s) (BS(s)) located upstream of a weak synthetic promoter in a way that the AD recruits RNAP to promote its binding to the promoter. As a result, the expression of the target gene is activated[26].

### Development of an arabinose-based toolkit for a wide range of transcriptional outputs in *Salmonella*. To maximize gene expression from $P_{BAD}$, it is necessary to identify conditions that shift the equilibrium from arabinose catabolism to maximal

induction of gene expression with minimal effect on cellular fitness[23,27]. The $P_{BAD}$ promoter has been extensively used to express heterologous and endogenous genes in *E. coli* and *Salmonella*. While the regulatory mechanism of the arabinose induction system has been well-characterized in *E. coli*[22,23,28], few studies have been performed that characterize the arabinose induction system in *Salmonella*[22,29]. In contrast to *E. coli*, *araE* is the only gene that encodes for an arabinose-specific importer in *Salmonella*[24,29]. In the native system, arabinose induces expression of the *araBAD* operon (which encodes three enzymes, AraB, AraA, and AraD, that convert arabinose to D-xylulose-5-phosphate to provide carbon and energy for cellular metabolism) and of the *araE* gene (Fig. 2a, Supplementary Data 1)[27].

Like in *E. coli*, where the addition of 0.2% arabinose results in maximal induction of $P_{BAD}$, 0.2% arabinose is also commonly used to induce both chromosomal and episomal $P_{BAD}$ in *S. enterica*[27]. In order to investigate the induction kinetics of the arabinose system in *Salmonella*, we quantified the output of a $P_{BAD}$-dependent fluorescent protein reporter after induction during mid-log growth phase in presence of various

concentrations of arabinose ranging from 0.00001% to 0.2% using flow cytometry (Fig. 2b, Supplementary Fig. 1, Supplementary Data 1). Our data demonstrate that the addition of 0.005% arabinose is sufficient to maximally induce $P_{BAD}$-controlled expression of plasmid-encoded superfolder green fluorescent protein (sfGFP) (Fig. 2b, light red) and 0.05% arabinose for the chromosomal construct (Fig. 2b, light grey). Deleting the *araBAD* operon, so that arabinose cannot be metabolized, resulted in markedly increased reporter protein expression compared to the *araBAD*[+] strain (Fig. 2b, dark grey and Supplementary Fig. 2a, dark dashed grey). A decoupled arabinose transporter-reporter system, where *araE* is constitutively expressed from the relatively weak $P_{proA}$ promoter[13], resulted in maximal induction at a 10-fold lower arabinose concentration than in the *araE*[+] strain, where *araE* is under control of its native (arabinose-inducible) promoter $P_{araE}$ (Fig. 2b, light grey and light red, 0.01% and 0.1%). A combination of decoupled arabinose transporter-reporter system and deletion of the *araBAD* operon resulted in increased $P_{BAD}$-dependent reporter gene expression over a wide range of 0.0005% to 0.2% arabinose (Fig. 2b, c, dark red, Supplementary Fig. 2a, dark dashed red, Supplementary Data 1 and 2). These results suggest that constitutive arabinose-independent, low-level expression of *araE* is optimal for an arabinose-inducible expression system in *S.enterica*.

A growth curve analysis further showed that in the *araBAD*[+] background in the presence of 0.2% arabinose, $P_{proA}$-controlled expression of *araE* ($P_{proA}$-*araE*) results in slower growth during early exponential growth (Fig. 2d, red, Supplementary Data 1), compared to native, arabinose-dependent *araE* expression in the wild-type strain ($P_{araE}$-*araE*) (Fig. 2d, grey). Upon entering later growth phases (from time point 330 min), the growth rate of the $P_{proA}$-*araE* strain exceeded that of the $P_{araE}$-*araE* strain. In contrast, strong constitutive *araE* expression from the $P_{proD}$ promoter resulted in similar growth compared to native arabinose-responsive *araE*-expression in later growth phases. In presence of 0.05% or lower arabinose concentrations, the growth of strains expressing *araE* from either native, $P_{proA}$, or $P_{proD}$ was similar (Supplementary Fig. 2b, Supplementary Fig. 3a, b, and c, Supplementary Data 3). In high arabinose concentrations (0.1% and 0.2%), the reduced growth of strains expressing *araE* from either native, $P_{proA}$, or $P_{proD}$ (in *araBAD*[+] background) is compensated by *araBAD* operon deletion (Supplementary Fig. 2c, d, e, Supplementary Fig. 3d, e, f, Supplementary Data 1 and 3). The high expression levels from the $P_{BAD}$-derived reporter system (Fig. 2b, c, and Supplementary Fig. 2a, highlighted in blue) with minimal growth defect (Supplementary Fig. 2b, c, and Supplementary Fig. 3, blue curves) implies an optimal 'arabinose induction window' ranging from arabinose concentrations of 0.01% to 0.1%.

**Arabinose-inducible CRISPR/dCas9-derived ATF library**. We next established an arabinose-inducible ATF system, which is based on a previously developed CRISPR/dCas9-derived ATF to activate gene expression in *E. coli*[17], and which employs a catalytically inactive version of *Streptococcus pyogenes* Cas9 (dCas9) as DBD[17] and a mutant version of SoxS as AD. The dCas9 binds to a scaffold RNA (scRNA), which consists of a sequence that targets the *J1*-derived synthetic promoter[17]. The scRNA is fused to an MS2 hairpin sequence, which recruits an MS2 coating protein (MCP) fused to an R93A mutant of the SoxS activator domain, which has previously been shown to effectively activate transcription[17,21]. (Fig. 3a, Supplementary Data 4). The dCas9 and SoxS(R93A) in this study were codon-optimized for expression in *Salmonella*.

In the ATF system established by Dong et al., dCas9 and scRNA expression are controlled by $P_{BAD}$, and SoxS is expressed from a constitutive promoter. We adapted this system in *S. enterica*, and used $P_{BAD}$ to control the expression of the relatively large dCas9 protein as well as the SoxS AD, to minimize the negative effect of CRISPR/dCas9-derived ATF expression on growth (Supplementary Fig. 4, Supplementary Data 5). The scRNA cassette, instead, was expressed from the strong constitutive $P_{BBa-\ J23119}$ promoter[30]. The dCas9 cassette was integrated into the chromosome at the attachment site of coliphage 186 (*att186*)[31]. Expression of the CRISPR/dCas9-derived ATF targeting region J105[17] upon the addition of 0.05% arabinose did not affect the growth of *S. enterica* before the mid-log phase (Supplementary Fig. 4). Therefore, we characterized our arabinose-inducible ATFs in mid-log phase cultures.

We selected crRNAs targeting sites located at 105, 106, 107, 108, or 111 bases upstream of the transcription start sites within the *J1*-derived synthetic promoter to establish the arabinose-inducible CRISPR/dCas9-derived ATFs in this study. These crRNAs have previously been shown to result in high transcriptional output in *E. coli*[17]. Moreover, to further extend the size of our library, we designed *J1* derivatives harboring two copies of J106 and J107 situated in a narrow region that is required for effective gene activation (J106-2x and J107-2x, respectively; for details, see the schematic shown in Supplementary Fig. 5). In *Salmonella*, CRISPR/dCas9-derived ATFs controlling expression of the fluorescent reporter protein resulted in minimal growth defect (Supplementary Fig. 6, Supplementary Data 6) and a high reporter expression after 4 h induction with arabinose (Fig. 3b and Supplementary Data 4, 16-fold induction for CRISPR/dCas9-derived ATF targeting J105 in M9 minimal medium, and Supplementary Fig. 7 and Supplementary Data 7, 2.5-fold induction for CRISPR/dCas9-derived ATF targeting J105 in LB medium).

In *E. coli*, a maximum of ~2.7-fold reporter expression was observed for CRISPR/dCas9-derived ATF after 4 h induction with 0.2% arabinose (Supplementary Fig. 8 and Supplementary Data 8, CRISPR/dCas9-derived ATF targeting J111). We were not successful in obtaining similar results for the CRISPR/dCas9-derived ATFs as reported by Dong et al., suggesting that gene activation is likely sensitive to the established expression system in *E. coli*. Further examination of the total population distribution showed reporter gene expression was induced in less than 36% of *E. coli* cells in the case of strong CRISPR/dCas9-derived ATF expression under the tested condition, *i.e.*, 0.2% arabinose (Supplementary Fig. 9). In contrast, more than 63% of *Salmonella* cells expressed reporter gene in the presence of 0.05% arabinose (Supplementary Fig. 10). As shown in Supplementary Fig. 9, the percentage of fluorescent *E. coli* cells increased to 71% after 8 h of induction (0.2 % arabinose). Therefore, we tested the inducible expression of CRISPR/dCas9-derived ATFs after 8 h induction and in the presence of various arabinose concentrations in *E. coli*, resulting in up to 7-fold induction (Fig. 3c, Supplementary Data 4). Additionally, we achieved fold improvement of induction using J106-2x and J107-2x promoters over the J106 and J107 promoters (J106-2x: 2.5-fold in both *E. coli* and *Salmonella*, and J107-2x: 2- and 3.5-fold in *E. coli* and *Salmonella*, respectively, Figs. 3b and 3c).

Stable gene expression is crucial for large-scale production. Therefore, we monitored the expression output of the CRISPR/dCas9-derived ATF targeting J105 in *Salmonella* over 36 h (Supplementary Fig. 11a, Supplementary Data 9). Fluorescent reporter expression was stable 12 h after induction of ATF targeting J105. To maintain the culture during fermentation, subculturing a microbial culture into a new growth medium is essential. To validate the expression stability in the microbial subculture, we measured the expression output of the strong

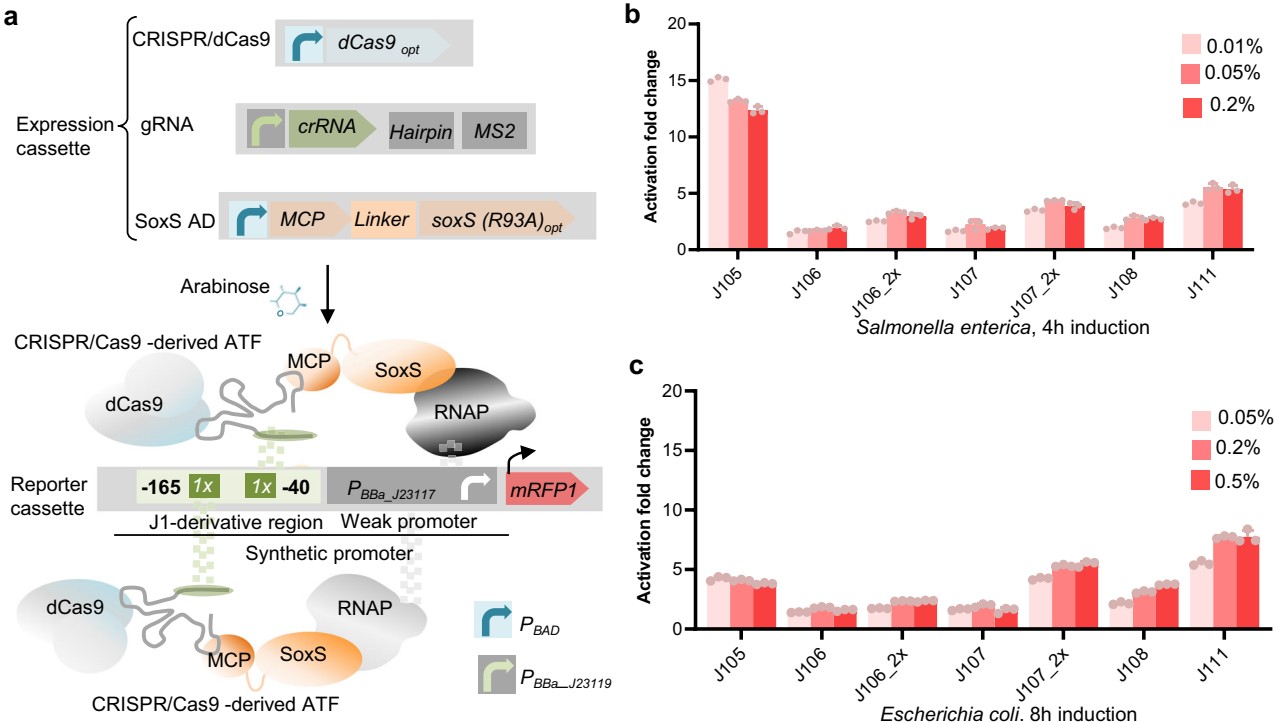

**Fig. 3 Arabinose-inducible CRISPR/dCas9-derived ATF library for tunable gene expression in Enterobacteriaceae. a** Schematic showing CRISPR/dCas9-derived ATFs developed in the present study. The expression cassette (top) of CRISPR/dCas9-derived ATFs comprises (i) CRISPR/dCas9 cassette containing a *dCas9* (codon-optimized for expression in *Salmonella enterica*) under control of $P_{BAD}$, (ii) gRNA cassette, comprising the crRNA fused to the MS2 RNA hairpin under control of the constitutive $P_{BBa-J23119}$[30], and (iii) AD cassette encoding MCP fused to SoxS(R93A) AD (codon-optimized for expression in *S. enterica*) via a 5aa-linker under control of $P_{BAD}$. In the presence of arabinose, dCas9 and SoxS are expressed. SoxS AD is recruited by MCP to the dCas9-scRNA complex via the MS2 RNA hairpin. The expressed ATF targets a *J1*-derived synthetic promoter[17] placed upstream of the gene of interest in the reporter cassette (bottom) via a sequence in the crRNA. The *J1*-derived synthetic promoter has potential crRNA target sites upstream of a weak $P_{BBa-J23117}$[17]. The schematic shows the condition in which the *J1*-derived synthetic promoter harbors two copies of the ATF BS, located between −165 and −40 bp upstream the TSS. The interaction of the ATF and RNAP induces the transcription of *mRFP1*[26]. Library of arabinose-inducible, CRISPR/dCas9-derived ATFs in *Salmonella* (**b**) and *E. coli* (**c**). Fluorescence fold induction relative to the non-inducing medium was measured in the presence of 0.005% (light red), 0.05% (red), and 0.2% (dark red) arabinose added to M9 minimal medium (supplemented 0.4% glycerol and 0.1% casamino acids) after 4 h for *S. enterica* and 8 h for *E. coli*. Data are expressed as the mean ± SD of the RFU obtained from three biological replicates, normalized to the $OD_{600}$. Asterisks indicate a statistically significant difference from the non-inducing medium (two-sided *t*-test; $*p \leq 0.05$, $***p \leq 0.001$). aa amino acid; AD activation domain; ATF artificial transcription factor; BS binding site; crRNA CRISPR RNA; DBD DNA binding domain; dCas9 catalytically inactive Cas9; gRNA guideRNA; MCP MS2 coat protein; mRFP1 monomeric red fluorescent protein 1; $OD_{600}$, optical density at 600 nm; RFU relative fluorescence units; RNAP RNA polymerase; SD standard deviation; TSS transcription start site. The full data are shown in Supplementary Data 4.

CRISPR/dCas9-derived ATF targeting J105 in *Salmonella* after three subsequent subcultures. We did not observe a significant difference in its induction level in subcultures (Supplementary Fig. 11b, Supplementary Data 9).

The inducible CRISPR/Cas9-derived ATFs allow tight control over transcriptional output with minimal basal expression through inducer concentration adjustment, diverse gRNA-encoding plasmids, and improved gRNA target sites in *Salmonella* and *E. coli*. The arabinose-inducible CRISPR/dCas9-derived ATF targeting the J105 site stands out as the most potent ATF, enabling exceptionally high enzyme expression in *Salmonella*. However, to maximize the performance of our arabinose-inducible CRISPR-derived ATFs in the *E. coli* expression system, additional optimizations should be considered, e.g., rewiring arabinose catabolism.

**Arabinose-inducible plant-derived ATF library.** An ideal regulatory toolkit should enable bioengineers to linearly induce gene expression over a wide dynamic range. Therefore, we aimed to extend the size of the library of arabinose-inducible ATFs for bioengineering applications in *Salmonella* and *E. coli*. Inspired by

MacDonald et al.[32], who demonstrated that the eukaryotic transcription factor QF from the fungus *Neurospora crassa* can be used in *E. coli* to activate transcription, we aimed to develop a new class of ATFs using heterologous TFs derived from plants for tunable gene expression in Enterobacteriaceae in this study. We previously showed that full-length plant-specific TFs ANAC102 and JUB1-derived ATFs function as weak and medium-strength ATFs in yeast[15]. In addition, the ATF derived from plant-specific GRF9 TF and the DBD of JUB1 TF ($JUB1_{DBD}$) function as medium-strength ATF in yeast[15]. Therefore, we selected candidate plant-derived TFs that may be suitable to function as ATFs covering a range of transcriptional outputs in Enterobacteriaceae.

In order to evaluate candidate plant-derived TFs, synthetic promoters harboring one, two, or four copies of the binding site of JUB1 (Supplementary Fig. 12a), ANAC102 (Supplementary Fig. 12c), or GRF9 (Supplementary Fig. 12c), were placed upstream of a fluorescent reporter protein in a low-copy plasmid (see Supplementary Information). Expression of plant-derived TFs under $P_{BAD}$-control in a high-copy plasmid (see Supplementary Information) induced with 0.2% arabinose was not sufficient to activate transcription in *E. coli* (Supplementary Fig. 13, Supplementary Data 10). Therefore, we fused the plant-derived

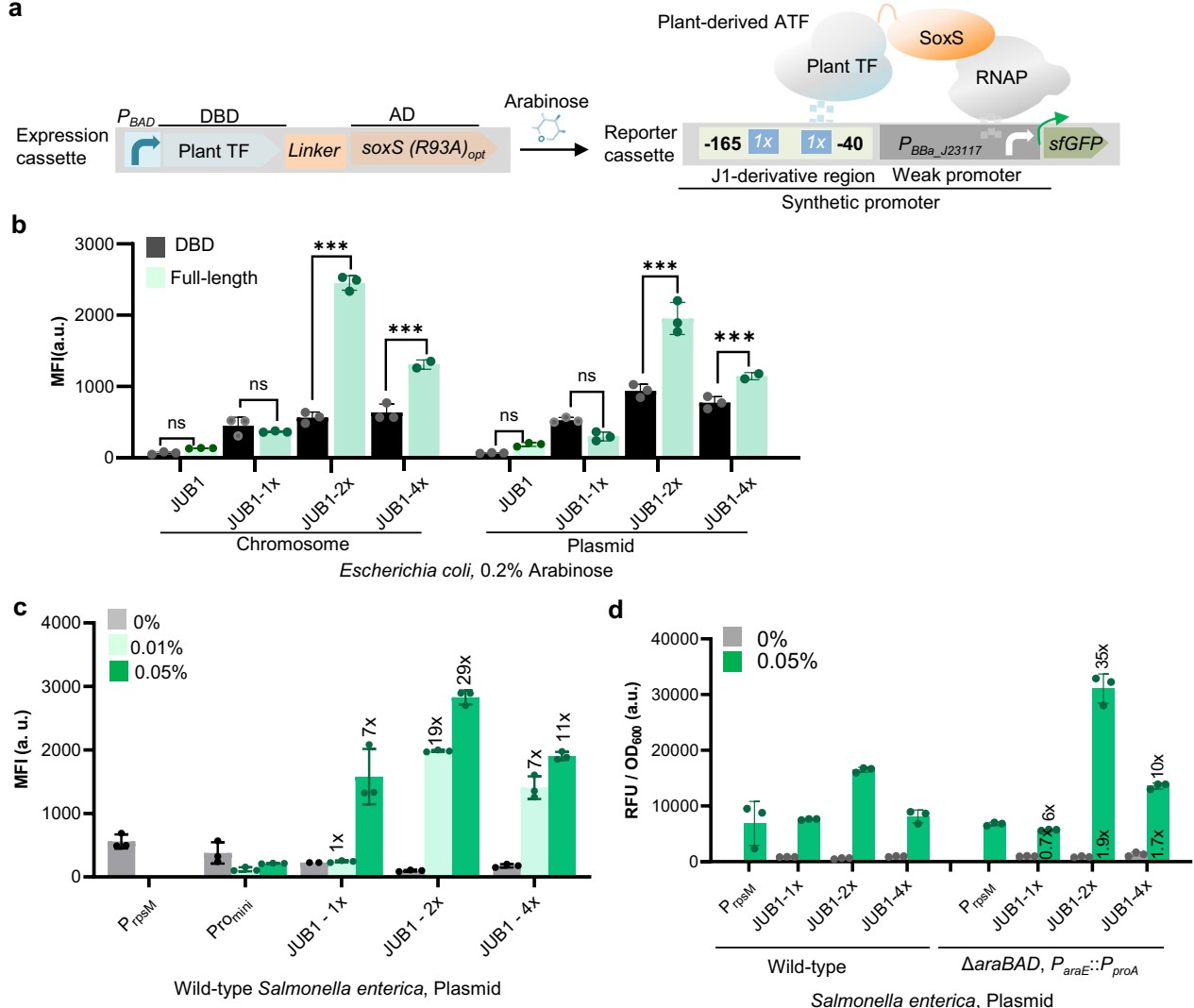

**Fig. 4 Arabinose-inducible plant-derived ATFs for tunable gene expression in Enterobacteriaceae. a** Schematic showing plant-derived ATFs developed in the present study. The expression cassette contains a plant TF fused via 5aa-linker to SoxS(R93A) activation domain expressed from a $P_{BAD}$ promoter. In the presence of arabinose, the expressed ATF binds a *J1*-derived synthetic promoter upstream of the weak $P_{BBa\_J23117}$[17]. In the schematic, a reporter cassette with a synthetic promoter harboring two copies of the ATF BS located between −165 and −40 bp upstream of the TSS is shown as an example. Interaction between the ATF and RNAP drives reporter sfGFP expression. **b** Arabinose-inducible JUB1-derived ATF in *Escherichia coli*. ATFs derived from full-length plant JUB1 or its DBD were chromosomally integrated at the *att186* site (black), or expression plasmids harboring JUB1- or only its DBD-derived ATF was transformed (green) into the cells with reporter cassette. Strains harboring the expression cassette, but no reporter cassette were used as a negative control. **c** Arabinose-inducible JUB1-derived ATF in *Salmonella*. Plasmids harboring full-length JUB1-derived ATF and reporter cassette were co-transformed into the *S. enterica* cell. The strain expressing the sfGFP from the constitutive promoter $P_{prsM}$ and strain harboring JUB1-derived ATF and reporter plasmid harboring minimal promoter were used as positive and negative controls, respectively. **d** Arabinose-inducible JUB1-derived ATFs in *Salmonella* with optimized arabinose catabolism and transport. The transactivation capacity of the JUB1-derived ATFs was tested in the wild-type background (green) and *S. enterica* strain with deleted *araBAD* and constitutively expressing *araE* from $P_{proA}$. The ATF transcription activation capacity in combination with one (1x), two (2x), or four (4x) copies of BS was tested in the non-inducing and inducing M9 minimal medium. For fold activation, the fluorescent output in the inducing medium was normalized to that in the non-inducing medium. Data are expressed as the mean ± SD of the MFI (**b**, **c**) or RFU (**d**) obtained from three biological replicates. "*x*" on top and inside the columns represent the fold induction compared to non-inducing medium and wild-type *Salmonella*, respectively. Asterisks indicate a statistically significant difference (two-sided *t*-test; ns, not significant; ∗$p \leq 0.05$; ∗∗$p \leq 0.01$; ∗∗∗$p \leq 0.001$). To simplify the figure, the ribosome binding site and terminator are not shown. M9 minimal medium was supplemented with 0.4% glycerol and 0.1% casamino acids. For induction, 0.2% arabinose was used in *E. coli* (8 h induction) and 0.0005% and 0.05% arabinose were used in *S. enterica* (4 h induction). aa amino acid; ATF artificial transcription factor; a. u. arbitrary units; BS binding site; JUB1 JUNGBRUNNEN1; MFI mean fluorescence intensity; RNAP RNA polymerase; RFU relative fluorescence units; sfGFP super-folder green fluorescent protein; TF transcription factor. The full data are shown in Supplementary Data 11.

TFs to the bacterial SoxS(R93A) activation domain and placed it under the control of $P_{BAD}$[17,21] (Fig. 4a, Supplementary Data 11). We evaluated the capacity of chromosomal or plasmid-based expression of these re-engineered plant-derived ATFs to activate transcription in *E. coli*. Our data indicate that an ATF based on

the full-length plant JUB1 TF, but not its DBD alone, exhibits a strong capacity to activate gene expression in *E. coli*. This is evident in Fig. 4b (0.05% arabinose added to the minimal medium and see Supplementary Data 11) and Supplementary Fig. 14 (0.2% arabinose added to LB medium and see

Supplementary Data 12), and aligns with our previous findings of the transcription activation capacity of ATFs in yeast[15].

Since we observed similar transcriptional outputs for both plasmid- and chromosomally-expressed systems, we implemented plasmid-based expression of JUB1-derived ATF in the following experiments, as it may be favorable over genomic integration in some cases due to easier accessibility to manipulation. The JUB1-derived ATF, in combination with one, two, or four copies of its binding site, was then characterized in *S. enterica*. As shown in Fig. 4c, in the minimal medium, the expression of the reporter (ATF expression cassette together with reporter cassette harboring corresponding binding site) displayed an inducible range of more than ~29-fold (for JUB1-2x) when induced with 0.05% arabinose, which is 40% higher than when grown in LB medium supplemented with 0.05% arabinose (Supplementary Fig. 15, Supplementary Data 13). Importantly, there was no significant correlation between the number of plant TF binding site copies in a given synthetic promoter and basal level of fluorescence expression (Supplementary Fig. 16, Supplementary Data 14), probably because the distance of the binding sites to the transcription start site (TSS) is out of the effective region to activate gene expression by the SoxS AD. Of note, all ATFs resulted in higher sfGFP reporter expression than the control strain expressing sfGFP from the constitutive $P_{rpsM}$ promoter (Fig. 4c and Supplementary Data 11, up to ~3.5-fold expression). All tested combinations of GRF9- or ANAC102-derived ATFs and their BSs resulted in low transcriptional outputs in *Salmonella* (Supplementary Fig. 17 and Supplementary Data 15, ~0.4- to ~0.6-fold of that observed for the control strain) despite being categorized as medium and strong transcription activators in yeast[15]. To further establish a dynamic arabinose-inducible system for precise temporal gene expression control in bacteria, we next characterized the JUB1-derived ATFs, together with one, two, or four copies of the BS, in a *S. enterica* background deleted for the *araBAD* operon (and, thus, deficient in arabinose catabolism), and constitutivly expressing arabinose transporter (Fig. 2b, $\Delta araBAD$, $P_{araE}::P_{proA}$). For the JUB1-2x expression/reporter in the optimized *Salmonella* genetic background, the addition of 0.05% arabinose resulted in ~1.9-fold induction of the reporter expression compared to constitutive expression and 35-fold induction compared to non-inducing conditions (Fig. 4d, Supplementary Data 11), with a minimal growth defect compared to the non-inducing conditions (Supplementary Fig. 18, Supplementary Data 16). The fluorescent expression output of strong JUB1-derived ATF targeting two copies of its binding site was stable over 24 h (Supplementary Fig. 19a, Supplementary Data 17). Additionally, we did not observe a significant difference in its induction level after three subcultures (Supplementary Fig. 19b, Supplementary Data 17). Here, we characterized the interplay between TF copy number (using plasmid- and chromosomal-based expression), the number of its target binding sites, and dose-response to the arabinose induction in wild-type or optimized genetic backgrounds modified for arabinose catabolism and transport. Our data suggest that arabinose-inducible JUB1-derived ATF, combined with two copies of its binding site, results in a high and stable expression level in *E. coli* and *Salmonella*.

**Plant-derived ATF to develop a *Salmonella* biosensor for alkaloid ligands**. A key obstacle in the microbial engineering of high-value chemicals is the quantification of chemical production[33]. d'Oelsnitz et al. developed a genetically encoded biosensor based on the multidrug-resistance repressor RamR from *S. enterica* to rapidly quantify the benzylisoquinoline alkaloid (BIA) group of plant therapeutic alkaloids[34]. They further

engineered *E. coli* for heterologous expression of RamR and demonstrated the utility of this sensor as a tool to detect BIAs, including the anti-tumor alkaloid noscapine (NOS), rotundine (ROTU), and tetra-hydro papaverine (THP)[35] by incorporating the RamR binding site into a synthetic promoter driving expression of a fluorescent reporter gene. In this reporter system, the presence of BIAs prevents the binding of the RamR repressor to its binding site, thereby triggering fluorescent reporter expression. Here, we developed a *Salmonella* strain that chromosomally expresses wild-type RamR under the control of its native promoter deleted for its RamR binding site and harbors a synthetic promoter with the RamR BS controlling sfGFP expression. We termed this strain SALSOR 0.1 (SALmonella biosenSOR 0.1, Fig. 5a and Supplementary Data 18). Low background signals in the absence of an alkaloid ligand generally lead to an increased signal-to-noise ratio and, thus, improved detection of the ligand (i.e., a lower detection limit). Therefore, we reasoned that employing the arabinose-inducible JUB1-derived ATF would allow us to increase the expression levels of the RamR repressor, and thus reduce the backgound signal. Therefore, we engineered the SALSOR 0.2 strain, which, in addition to harboring the synthetic RamR-controlled sfGFP reporter module, expresses chromosomally encoded RamR under the control of our above-developed arabinose-inducible JUB1-derived ATF in combination with a synthetic promoter harboring two copies of the JUB1 BS (JUB1-2x) (Fig. 5b, Supplementary Data 18). The established two-input SALSOR 0.2 biosensor requires arabinose to repress fluorescent protein expression that can be restarted by adding a BIA ligand. Using SALSOR 0.1, we observed ~1.2-, 1.3-, and 1.8-fold fluorescent signal over background noise (signal-to-noise ratio) for NOS, ROTU, and THP ligands in the presence of arabinose (Fig. 5a). SALSOR 0.1 displayed the limitation to detect NOS ligand that shows weak binding affinity the biosensor, consistent with the observations of d'Oelsnitz et al.[34]. As shown in Fig. 5b, the detection of alkaloid ligands by the SALSOR 0.2 biosensor strain leads to a significantly higher signal-to-noise ratio compared to SALSOR 0.1. In the absence of arabinose, there is no ATF to stimulate RamR expression. As a result, the RamR repressor is not expressed, and the expression level of sfGFP that is situated after a promoter containing a binding site for RamR significantly increases (Fig. 5b). Thus, using the two-input SALSOR 0.2 biosensor allows improved detection of the interacting ligand compared to a one-input biosensor, such as SALSOR 0.1, as the system's output can be adjusted by applying different concentrations of both inducer and ligand. This is particularly useful in metabolic engineering, where precise detection of a ligand is essential.

**Arabinose-inducible plant-derived ATF to control β-carotene production in *E. coli***. We next aimed to validate the capacity of plant-derived ATFs to activate the transcription of large operons for metabolic engineering applications in Enterobacteriaceae. To this end, we chose to evaluate the transcriptional control of the well-characterized β-carotene biosynthesis pathway using plant-derived ATFs. To convert bacterial farnesyl pyrophosphate (FPP) precursor to β-carotene, geranylgeranyl diphosphate synthase (CrtE), phytoene synthase (CtrB), phytoene desaturase (CrtI) and lycopene cyclase (CrtY) enzymes are needed (Fig. 6a). We first transformed *E. coli* with the pK2151201 plasmid (constructed by Glasgow iGEM 2016, Fig. 6b) harboring the *crtE*, *crtB*, *crtI*, and *crtY* genes in a single operon placed under control of the constitutive, strong promoter $P_{J23106}$ (designed by Anderson et al., iGEM2006), with each gene placed downstream of an individual RBS. We next implemented a synthetic promoter containing five copies of the JUB1 TF binding site to control the expression of the

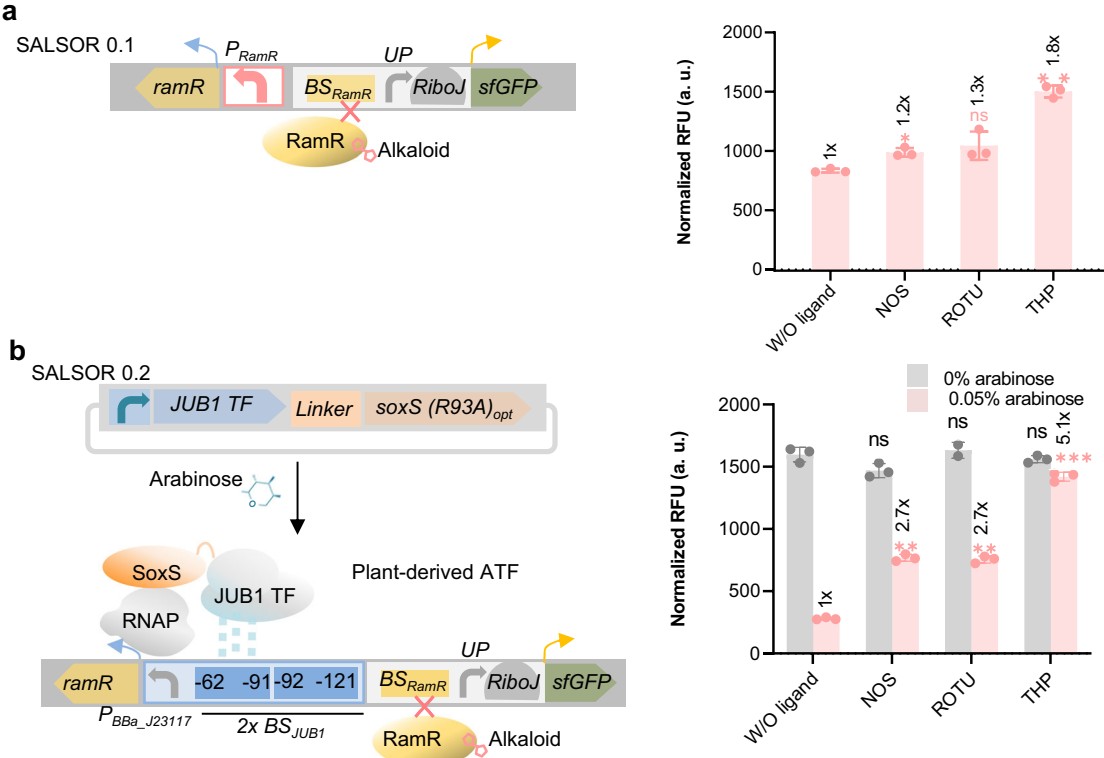

**Fig. 5 *Salmonella* biosensor responsive to alkaloids. a** Fluorescence response of SALSOR 0.1 to alkaloids. The RamR sensor's native constitutive promoter deleted for the RamR binding site (present in *Salmonella enterica*) was used to express RamR. Expressed RamR targets a synthetic promoter that controls sfGFP expression. The synthetic promotor consists of a RamR BS ($BS_{RamR}$) and an *UP* element, followed by a RiboJ-RBS (75 bases consisting of a satellite RNA of tobacco ringspot virus-derived ribozyme followed by a 23-nucleotide hairpin immediately downstream to help expose the RBS)[34,52]. In the absence of alkaloids, RamR represses reporter expression. In the presence of alkaloids, RamR interaction with $BS_{RamR}$ is inhibited, and reporter expression is induced. The fluorescence output of SALSOR 0.1 was measured in LB medium in the absence and presence of alkaloid ligands NOS, ROTU, and THP. **b** Fluorescence response of SALSOR 0.2 to alkaloids. A plasmid encoding JUB1-derived ATF was transformed into S. *enterica* optimized for arabinose catabolism and its transport($\Delta araBAD$, $P_{araE}::P_{proA}$). A synthetic promoter containing two copies of the JUB1 BS fused to the weak $P_{BBa\_J23117}$[17] was used to express the RamR sensor. The expressed RamR targets the synthetic promoter located upstream of *sfGFP*. The fluorescence output of SALSOR 0.2 was measured in the absence and presence of alkaloid ligands in non-inducing (grey) and inducing (0.05% arabinose; pink) M9 minimal medium supplemented 0.4% glycerol and 0.1% casamino acids after 4 h. The maximum ligand concentrations of NOS (100 µM), ROTU (250 µM), and THP (1 mM) based on the compound's solubility limit in 1% DMSO were used. Data are expressed as the mean ± SD of the RFU obtained from three biological replicates and are normalized to that of WT treated with DMSO without any ligand. "x" on top of the columns represents the fold induction compared to without ligand. Asterisks and "ns" indicate the statistical significance of the difference to the inducing and non-inducing medium, respectively (two-sided *t*-test; ns, not significant; ∗$p \le 0.05$; ∗∗$p \le 0.01$; ∗∗∗$p \le 0.001$). To simplify the figure, the RBS upstream of the ATF and terminators are not shown. AD activation domain; ATF artificial transcription factor; a. u. arbitrary units; BS binding site; JUB1 JUNGBRUNNEN1; NOS noscapine; $OD_{600}$, optical density at 600 nm; RBS ribosome binding site; RNAP RNA polymerase; RFU relative fluorescent units; ROTU rotundine; SD standard deviation; sfGFP superfolder green fluorescent protein; THP tetra-hydropapaverine; TSS transcription start site; UP upstream element; W/O without; WT wild-type. The full data are shown in Supplementary Data 18.

*crtEBIY* operon in pCAROTENE5X and integrated the arabinose-inducible JUB1-derived ATF into the *att186* site of *E. coli*. The pK2151200 plasmid harboring the *crtEBIY* operon without a promoter (constructed by Glasgow iGEM 2016, Fig. 6c) was used as a control.

As shown in Fig. 6d (see also Supplementary Data 19), in the presence of 0.2% arabinose, the strain with ATF control modules produced ~2.1-fold more β-carotene compared to the strain harboring pK2151201. Increasing the induction time from 4 h to 8 h led to a ~ 40% increase in β-carotene production in the strain allowing expression of the biosynthetic pathway genes using ATF (Fig. 6e, Supplementary Fig. 20, Supplementary Data 19 and 20, $ATF_{JUB1}$, pCAROTEN5X, 0.2% arabinose). However, it did not result in an increased β-carotene level in the strain with constitutive operon expression. The β-carotene production was stable for at least 12 h (Fig. 6e, Supplementary Data 19). These data suggest a high capacity of the arabinose-inducible JUB1-

derived ATF to increase the transcription level of the *crtEBIY* operon and β-carotene production compared to expression from the constitutive promoter (Fig. 6d).

## Discussion

To better understand the regulatory mechanisms underlying Gram-negative pathogenesis as well as to implement these bacteria for bioengineering applications, species-specific orthogonal, programmable expression systems are required. The native promoters of Gram-negative bacteria have been used to control the expression of target genes. However, native bacterial promoters are under the control of endogenous bacterial TFs, which may interfere with the endogenous regulatory networks of the host, making them suboptimal for synthetic biology applications[14]. Furthermore, the most effective production strategy preferably combines a biomass growth phase followed by a protein-of-

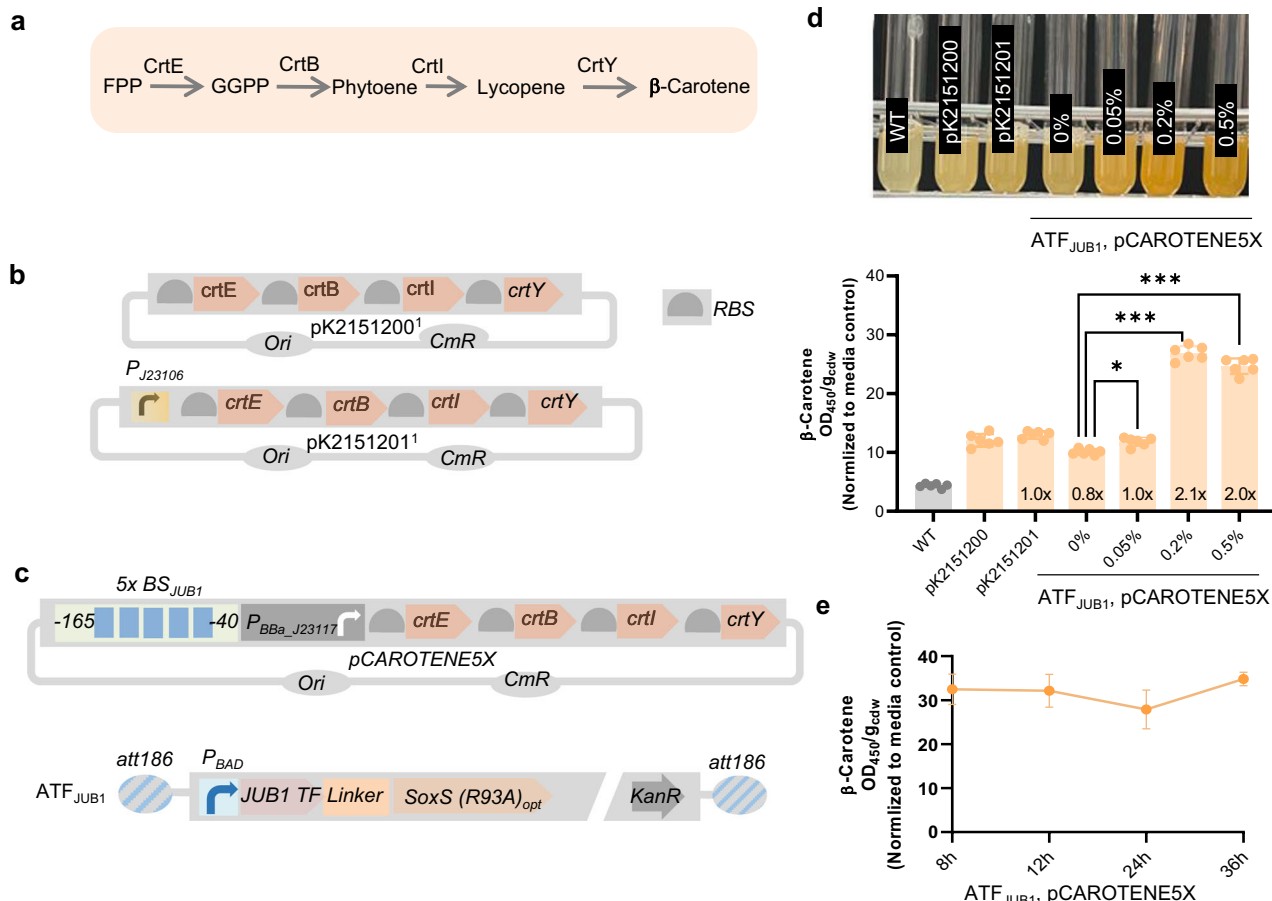

**Fig. 6 Production of β-carotene in *E. coli* strains. a** Schematic showing the bacterial β-carotene production from FPP. CrtB, phytoene synthase; CrtY, lycopene cyclase; CrtI, phytoene desaturase; CrtE, geranylgeranyl pyrophosphate synthase. **b** Schematic showing the control β-carotene-encoding plasmids used in this study. In pK2151200, BioBricks K118014 (RBS+*crtE*), K118006 (RBS+*crtB*), K118005 (RBS+*crtI*), and K118013 (*crtY*) are placed in a single operon, *crtEBIY* (constructed by Glasgow iGEM 2016). In plasmid pK2151201, bacterial constitutive strong promoter *J23106* (designed by J. Anderson, iGEM2006) controls the expression of the synthetic *crtEBIY* operon. **c** Schematic representing the production of β-carotene controlled by the JUB1-derived ATF. In plasmid pCAROTENE5X, a *J1*-derived synthetic promoter containing five copies of the JUB1 TF binding site fused to $P_{BBa\_J23117}$[17] controls the expression of the *crtEBIY* operon. A JUB1-derived ATF cassette contains the full length of JUB1 TF linked via a 5aa linker to SoxS(R39A)[17] under control of $P_{BAD}$. It additionally encodes KanR. The cassette is flanked by 40-bp homology arms to integrate into the *att186* site[31] in the *E coli* chromosome. To simplify the figure, the RBS upstream of ATF and terminators are not shown. **d** Analysis of carotenoid content of *E. coli* strains. The JUB1-derived ATF was integrated into the *att186* site of *Escherichia coli*. Followed by pCAROTEN5X transformation, β-carotene production was quantified after 8 h growth in the absence and presence of 0.05%, 0.2%, and 0.5% arabinose added to the LB medium (ATF_JUB1, pCAROTENE5X). Representative cultures of the β-carotene-producing strains are shown at the top. **e** Time course of carotenoid content controlled by JUB1-derived ATF in *E. coli*. For the *E. coli* strain harboring the JUB1-derived ATF integrated into the *att186* site and transformed with pCAROTEN5X, the carotenoid content was measured after 8 h, 12 h, 24 h, and 36 h, in the presence of 0.2% arabinose added to the LB medium. The β-carotene absorbance at 450 nm was measured using the method reported by Lian et al.[50], divided by cdw. The values were normalized to that of the LB medium. Values represent the mean ± SD of three independent colonies. *E. coli* strains containing pK2151200 and pK2151201 were used as controls. Asterisks indicate a statistically significant difference (two-sided *t*-test; ns, not significant; ∗∗∗$p ≤ 0.001$). "*x*" inside the columns represents the fold induction compared to pK2151200. aa amino acid; AD activation domain; ATF artificial transcription factor; BS binding site; cdw cell dry weight, *cmR* chloramphenicol resistance gene; FPP farnesyl pyrophosphate; GGPP geranylgeranyl pyrophosphate; JUB1 JUNGBRUNNEN1; KanR kanamycin resistance marker; ori origin of replication; RBS ribosome binding site; TF transcription factor; WT wild type. Full data are shown in Supplementary Data 19.

interest production phase[23]. However, orthogonal inducible gene expression systems are largely absent from the methodological toolbox currently employed in prokaryotic synthetic biology research, highlighting the need for further development in this area[36–38].

In this study, we evaluated the usage of arabinose-inducible ATFs using CRISPR/dCas9- or plant-derived TFs to control gene expression in Gram-negative bacteria. To demonstrate a practical application of the plant-derived ATFs developed in this study, we employed JUB1-derived ATF for β-carotene production in *E. coli*. An *E. coli* strain harboring the β-carotene biosynthesis operon

placed under the control of JUB1-derived ATF produced 2.1-fold higher levels of β-carotene compared to a strain expressing the β-carotene biosynthesis operon from a constitutive promoter, opening up a new door for applying plant-derived ATFs to regulate metabolite production in prokaryotic microbial cell factories. We further demonstrated that plant-derived ATFs are a promising tool for developing *Salmonella*-based biosensors. Here, we engineered the SALSOR 0.2 biosensor that is able to sensitively detect the alkaloids NOS, ROTU, and THP. The SALSOR 0.2 strain might be used for high-throughput screening of other alkaloid products in chemical engineering projects. The complete

microbial biosynthesis of NOS has recently been reported[39], and, therefore, it might be a valuable potential anticancer metabolite to be sustainably manufactured in bacterial cell factories. An interesting potential application of SALSOR 0.2 strain could be its application as the whole-cell biosensor for rapid quantification of the extracellular alkaloid concentration produced by other microbial cell factories[33,40]. Additionally, the plant-derived ATFs can be employed to establish sensitive biosensors to detect bicyclic monoterpenes alkaloids[41], other natural products[42], or heavy metals[43–45].

The generated ATFs can be adapted to different induction systems, including chemical inducers (e.g., tetracycline, IPTG) as well as optogenetic inducers[46]. By placing a single inducible promoter upstream of the ATFs, we should be able to express multiple proteins at different levels, which is crucial for bioengineering projects aimed at engineering lengthy biosynthetic pathways or studying complex gene regulatory networks. However, to successfully implement a new inducer to activate ATF expression, the chemical induction system requires to be adequately characterized. Thus, our ATFs offer promising alternatives to the currently available expression systems, such as the T7 RNA polymerase-based expression system. Furthermore, the ATF transcriptional output might be further fine-tuned using different gRNAs (for CRISPR/dCas9) or by altering the copy numbers of plant-derived TF binding sites. We note that more than 2000 plant-specific TFs exist that are evolutionarily distant from bacteria, highlighting the potential for future ATF developments[47]. In situations where very low or high expression of biosynthetic pathway genes for complex metabolites may lead to failed compound production or production at toxic levels[18], our collection of bacterial ATFs, spanning weak to strong transcriptional outputs, offers a promising solution. Finally, our collection of arabinose-inducible ATFs, allowing precise control of gene expression in Enterobacteriaceae, may offer valuable tools to tune gene expression in other Gram-negative bacteria, such as *P. putida*, *P. aeruginosa*, and *Cyanobacteria*.

## Methods

**General**. Strains used in this study derive from *S. enterica* serovar Typhimurium strain LT2, *E. coli* DH10β (NEB, Frankfurt am Main, Germany), and *E. coli* DH10B-ALT (*E. coli* DH10B modified to constitutively express *araC*, Addgene, #61151) and are listed in Supplementary Table 1. Plasmids were constructed using the NEBuilder HiFi DNA assembly strategy of New England Biolabs (NEB, Frankfurt am Main, Germany)[48], or digestion and ligation using T4-DNA ligase (NEB, Frankfurt am Main, Germany). Plasmid and primer sequences are listed in Tables S2 and S3, respectively. PCR amplification of DNA fragments was performed using high-fidelity polymerases: Q5 DNA Polymerase (New England Biolabs, Frankfurt am Main, Germany), Phusion Polymerase (Thermo Fisher Scientific), or PrimeSTAR GXL DNA Polymerase (Takara Bio, Saint-Germain-en-Laye, France) according to the manufacturers' recommendations. All restriction enzymes were purchased from New England Biolabs (Frankfurt am Main, Germany). Amplified and digested DNA fragments were gel-purified prior to further use. Primers were ordered from IDT (Integrated DNA Technologies Inc., Dessau-Rosslau, Germany) and Sigma-Aldrich (Deisenhofen, Germany). All gBlocks were ordered from IDT (Dessau-Rosslau, Germany). Standard *E. coli* cloning strains were NEB dam⁻/dcm⁻, NEB 5α, or NEB 10β (New England Biolabs) were transformed by heat-shock to propagate constructed plasmids. Strains were grown in Luria-Bertani (LB) medium containing 10 g Tryptone, 5 g Yeast Extract, 5 g NaCl per liter, and an appropriate antibiotic(s) (Ampicillin, 100 µg/ml; Chloramphenicol, 50 µg/ml; or Kanamycin, 50 µg/ml).

The integrity of plasmid constructs was confirmed by sequencing (Microsynth Seqlab, Goettingen, Germany).

Plasmids or integration fragments were amplified by PCR, and using λ red-mediated homologous recombination standard protocols, they were next integrated into the chromosome of *S. enterica* strain LT2 (see Supplementary Information), or *E. coli* (NEB). Integrations into chromosomal *attB* sites of *S. enterica* strain LT2 or *E. coli* DH10B-ALT target strains were performed as described by St-Pierre et al.[2]. DNA fragments were introduced by electroporation into the cells. Chromosomal integrants were selected using the appropriate antibiotic(s) (Ampicillin, 50 µg/ml; Chloramphenicol, 15 µg/ml; or/and Kanamycin, 25 µg/ml). When required, an appropriate concentration of arabinose (see Results) was added. Single copy integration of each linearized fragment into the target chromosomal site was verified by colony PCR (and sequencing). Methodologies to construct the plasmids and strains to characterize the arabinose toolbox and its ATFs are described in Supplementary Information.

**Biosensor strain**. To establish SALSOR 0.1, a 136-bp fragment between RamR (STM580) and its responsive promoter, controlling RamA (STM581) expression in *S.* Typhimurium LT2 (NC003197, gift from Kelly T. Hughes) was replaced with PCR-amplified *KanSceI* cassette (primer pair RamR-Kan-fv/RamR-Kan-rv, on pWRG717[49]) using λ red-mediated homologous recombination standard protocols. The generated strain was called RKLT2. The "SELEX alkaloid sensor" containing the native promoter of *ramR* and its RBS (to deliver RamR expression) and RamR-responsive operator from the promoter of *ramA* (to be targeted by RamR) fused to synthetic minimal promoter region (UP element and RiboJ RBS), sfGFP and rrnBT1 terminator was designed and synthesized by ATG bioscience. Next, the "KanSceI" cassette of RKLT2 was replaced with a PCR-amplified "SELEX alkaloid sensor" cassette (primer pair RamR-Sensor-fv/RamR-Sensor-rv, on "SELEX alkaloid sensor") using λ-RED-mediated homologous recombination standard protocols. The generated strain was called SALSOR 0.1. To establish two-input alkaloid biosensor SALSOR 0.2 strain, a 136-bp fragment between RamR (STM580) and its responsive promoter, controlling RamA (STM581) expression in EM12877 (generated in this study) was replaced with PCR-amplified *KanSceI* cassette (primer pair RamR-Kan-fv/RamR-Kan-rv, on pWRG717[49]) using λ red-mediated homologous recombination standard protocols. The generated strain was called RKLT3. The *KanSceI* cassette of RKLT3 was replaced with the "ATF alkaloid sensor" cassette, containing a synthetic promoter with four copies of JUB1 binding site and synthetic RBS (to deliver RamR expression) and RamR-responsive operator from the promoter of *ramA* (to be targeted by RamR) fused to synthetic minimal promoter region (UP element and RiboJ RBS), sfGFP, and rrnBT1 terminator was PCR-amplified from synthetic fragment synthesized by ATG bioscience (primer pair RamR-Sensor-fv/RamR-Sensor-rv) using λ red-mediated homologous recombination standard protocols. The generated strain was called SALSOR4X. Next, plasmid pJUB1 was introduced into SALSOR4X via electroporation to generate SALSOR 0.2 strain.

**β-Carotene-encoding plasmids**. To construct pCAROTENE5X, the origin of replication, Ampicillin resistance gene, and synthetic promoter harboring five copies of JUB1 BS were PCR-amplified from pJUB1-5X (see Supplementary Information) using primer pair 5X-fv and 5X-rv. Fragments K118014 (RBS+crtE), K118006 (RBS+crtB), K118005 (RBS+crtI) and K118013 (crtY) were PCR-amplified from pK2151200 (BBa_K2151200 assembled into pSB1C3, gifted by S. Colloms) using primer pair K2151200-fv and K2151200-rv. The two PCR fragments were assembled using the NEBuilder HiFi DNA assembly strategy to generate "pCAROTENE5X".

Induction experiments. Single colonies of bacterial reporter strains were inoculated into 2 mL LB or M9 minimal medium supplemented with appropriate antibiotics and grown at 37 °C (*E. coli*) or 30 °C (*S. enterica* strain LT2), 220 RPM overnight. The M9 minimal medium contains 12.8 g $Na_2HPO_4.7H_2O$, 3 g $K_2HPO_4$, 0.5 g NaCl, 0.5 g MgSO4.7H2O, 1 g $NH_4Cl$, 0.34 g thiamine hydrochloride, 0.4% glycerol, and 0.1% casamino acids per liter. Late stationary phase cultures were diluted 1:100 in media supplemented with appropriate antibiotic(s). For inducible system construction with $P_{BAD}$, strains were inoculated in a medium supplemented with appropriate antibiotic(s) and arabinose at 0.0001%, 0.0005%, 0.001%, 0.005%, 0.01%, 0.05%, 0.1%, or 0.2% and cells were harvested at a certain time point (see Results).

**Plate reader experiments**. Cells were inoculated in a 2 mL medium supplemented with appropriate antibiotics for fluorescent reporter experiments and grown at 30 °C (*S. enterica*), or 37 °C (*E. coli*), 220 RPM overnight. Overnight cultures were diluted 1:100 in fresh medium with appropriate antibiotic(s) (see "Supplementary Method"), and appropriate concentration of arabinose (see Results) and 150 µL were aliquoted in triplicate into flat, clear-bottomed 96-well black plates (Greiner Bio-one). Plates were closed with a sterile lid and condensation rings (Greiner Bio-one), and incubated with shaking at 30 °C (*S. enterica*), or 37 °C (*E. coli*) in a Biotek Synergy H1 plate reader. $OD_{600}$ and mRFP1 fluorescence (excitation 510 ± 10 nm, emission 625 ± 20 nm), or $OD_{600}$ and sfGFP fluorescence (excitation 478 ± 10 nm, emission 515 ± 20 nm) were measured every 10 min. For ATF library characterization (Fig. 3c and Fig. 4d), the fluorescence measured divided by the $OD_{600}$ was reported.

For biosensor characterization in *Salmonella* strains, we adapted the protocol reported by d'Oelsnitz et al.[34] for biosensor characterization in *E. coli*. The cells were inoculated in 2 mL M9 minimal medium supplemented with appropriate antibiotics and grown at 30 °C, 220 RPM overnight. The following day, 20 µl of each culture was then used to inoculate six separate cultures in a 2 mL 24 Deep Well RB Block (Thermo Scientific) containing 900 µl medium, three for test alkaloid ligands in the presence of 0.05% arabinose and three for the alkaloids in the absence of arabinose. The Deep well was closed with a Seal. For the control experiment, the alkaloid was not to the non-inducing and inducing medium (M9 minimal without and with 0.05% arabinose). After 3.5 h of growth at 30 °C, cultures were induced with 100 µl medium containing either 10 µl DMSO or 100 µl medium containing the target alkaloid dissolved in 10 µl DMSO. The maximum ligand recommended concentrations (100 µM, 250 µM, and 1 mM for NOS (Merck, Frankfurt, Germany), ROTU (Absource, Munich, Germany), and THP (BIOZOL, Eching, Germany), respectively) was used based on the compound's solubility limit in 1% DMSO. Cultures were grown for an additional 7 h at 37 °C and 250 RPM and subsequently centrifuged (3500 g, 4 °C, 10 min). The supernatant was removed, and cell pellets were resuspended in 1 ml PBS (137 mM NaCl, 2.7 mM KCl, 10 mM $Na_2HPO_4$, 1.8 mM $KH_2PO_4$, pH 7.4). 100 µL of the cell resuspension for each condition was transferred to a 96-well microtiter sterile black, clear bottom plate closed with a lid (Greiner Bio-One) to perform the plate reader experiment.

**Flow cytometry and data analysis**. For fluorescent reporter experiments, cells were inoculated in 2 mL LB or M9 minimal medium supplemented with appropriate antibiotics and grown at 30 °C, 220 RPM overnight. Overnight cultures were diluted 1:100 in fresh 2 mL medium with appropriate antibiotics (see

"*Construction of plasmids and strains*") and appropriate concentrations of arabinose (see Results) and grown at 37 °C (*S. enterica*), 220 RPM. The cultures were harvested at the desired time point(s) after induction (see Results), diluted 1:40 in PBS, and analyzed using an SH800S cell sorter Flow Cytometer (Sony). sfGFP fluorescence values were obtained from a minimum of 10,000 cells in each sample. The mean GFP fluorescence per cell was calculated using FlowJo Software. For optimization of the arabinose-based toolkit in *Salmonella* (Fig. 2b), cells were inoculated in 2 mL LB supplemented. Ampicillin was added to the cultures for plasmid-based expression. The cultures were grown at 37 °C, 220 RPM overnight. Overnight cultures were diluted 1:100 in fresh 2 mL LB (Ampicillin for plasmid-based expression was used), and different concentrations of arabinose were added to the cultures and grown at 37 °C, 220 RPM for 150 min (chromosome-based expression) or 180 min (plasmid-based expression). Next, the $OD_{600}$ was measured, and cells were fixed in a 4% paraformaldehyde (PFA) solution. Briefly, the cultures were centrifuged, the supernatants were discarded, and the pellets were resuspended in 500 µL of 4 % PFA. After incubation for at least 5 min at room temperature, the cells were washed with PBS before the flow cytometry analysis.

**β-Carotene production and quantification**. The *E. coli* strain 10β (NEB) transformed with pK2151200 or pK2151201 plasmids were plated on LB, supplemented with chloramphenicol. The *E. coli* strains genetically modified at the *att186* site for arabinose-dependent induction of JUB1-derived ATF and transformed with pCAROTENE5X plasmid were plated on LB, supplemented with kanamycin and chloramphenicol. Cells were grown at 30 °C for 24 h. The colonies were inoculated into a 4 mL non-inducing LB medium and grown overnight at 30 °C and 200 RPM in a rotary shaker. A day after, the pre-cultures were used to inoculate the main cultures (4 ml) in an inducing LB medium, i.e., containing 0.2% arabinose. All cultures were inoculated from pre-cultures to an initial $OD_{600}$ of 0.1. Cells were grown for 24 h at 30 °C and 200 RPM to saturation.

Stationary phase bacterial cells were collected by centrifugation at 13,000×g for 1 min and, using the method reported by Lian et al.[50], β-carotene concentration was assessed. Briefly, cell pellets were resuspended in 1 ml of 3 M HCl, boiled for 5 min, and cooled in an ice bath for 5 min. Next, the lysed cells were washed with double distilled $H_2O$ and resuspended in 400 µl acetone to extract β-carotene. The cell debris was removed by centrifugation. The extraction step was repeated until the cell pellet appeared white. The β-carotene-containing supernatant was analyzed for its absorbance at 450 nm ($A_{450}$). The production of β-carotene was normalized to the cell density.

**Reporting summary**. Further information on research design is available in the Nature Portfolio Reporting Summary linked to this article.

## Data availability

The relevant data are available from the corresponding authors upon request. The source data underlying Figs. 2b–e, 3a–c, 4b–d, 5a–b, 6d-e, Supplementary Figs. 2a–c, 3a–f, 4, 6a–g, 7, 8, 9 10, 11a–b, 14, 15, 16, 17, 18a–c, 19a–b, and 20 are provided as a Supplementary Data files. The main expression and reporter plasmids constructed in this study are available from Addgene (www.addgene.org) under the following ID numbers: pSJ105, 208859; pSJ106-2x, 208860; pSJ107-2x, 208812; pJUB1, 208813; pJUB2x, 208814; pG0A0-1-1-PBAD-dCas9, 208815; pCAROTENE5X, 208816.

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

## Acknowledgements

We thank Sean Colloms (University of Glasgow) for providing the plasmids pK2151200 and pK2151201, and Kelly T. Hughes (University of Utah) for kindly providing the *Salmonella* strains TH437, TH3730, and TH6701. This work was supported in part by the European Research Council (ERC) under the European Union's Horizon 2020 research and innovation program (grant to M.E.; agreement no. 864971). E.C. (Max Planck Scientific Member) and M.E. (Max Planck Fellow) acknowledge the Max Planck Society for its support. E. C. also thanks the German Research Foundation (support through the Leibniz Prize).

## Author contributions

M.E. and G.N. conceived the study. G.N. designed expriments. G.N. and H.R. designed arabinose induction experiments. G.N. analyzed the data with a contribution from H.R.; G.N. and M.E. wrote the manuscript. E.C. and M.E. contributed funding and resources. All authors proof-read and approved the manuscript.

## Funding

## Competing interests

The authors declare no competing interests.
