## [Peer Review File · Communications Biology]

Reviewers' comments:

Reviewer #1 (Remarks to the Author):

The revised manuscript provided by the authors represents a significant improvement over the original version, containing additional figures and explanations that address most of this reviewer's concerns. Notably, the authors have improved the maximum induction ratio of their aTF to 35-fold, sufficiently demonstrated tunable expression using their aTFs, and improved the performance of the alkaloid biosensors and carotenoid production strains. The text edits made to this revision help to clarify the novelty of the plant-based aTFs, better support the claims, and improve the accuracy of the manuscript as a whole. Overall, this work should be of interest to the microbial synthetic biology community.

The following minor edits are suggested to optionally improve the clarity of the manuscript.

1. Using the more common abbreviation THP for TetraHydro Papaverine, instead of TPH used throughout this manuscript, may improve clarity.
2. In the legend of figure 4, subpanel "e" is referenced, but does not exist. Should this instead be subpanel "d"?
3. The language in the discussion (page 17, lines 543-555) should be toned down where appropriate. Statements such as "The generated ATFs can be adapted to different induction systems, including chemical inducers (e.g., tetracycline, IPTG) as well as optogenetic inducers" should instead indicate that the aTFs may prove possible to adapt, since the authors did not directly test the aTFs with these inducers.

Reviewer #3 (Remarks to the Author):

This MS explains the construction of several arabinose inducible expression systems based on dCas9 and plant-derived TF. Then, they validate those systems with two examples: i) building a biosensor strain and ii) constructing a strain to produce b-carotene. In general terms the paper and the developed tools are interesting (even though they would not be my first choice) but it would be great to have a revised version of the MS addressing the following points: detailed and focused content and clearer conclusions.

Below there are some comments and questions that need to be addressed before publication:

I would change the term "Gram-negative" in the title since this technology is only appropriate for just a limited number of gram-negatives, Enterobacteria.

The intro section should be shortened by going directly to the point! The necessity of expression systems to understand GRNs or to create biotech applications or for metabolic engineering.

What about the Anderson constitutive promoter library (doi:10.1186/1754-1611-3-4)?

What about commonly used expression system? Why are not mentioned in the MS?

How stable are these constructs? It seems easy (many regulators involved) to get spontaneous mutant that STOP the entire expression system. Would that explain the low percentage of cells expressing the reporter?

P3L102. If we want to use this system, do we have to engineer the L-ara pathway? And what exactly mean that?

Would this ara system would work in other Gram-negative bacteria?

P7L229. Does this dCas9-SoxS Salmonella codon optimized results in a less optimal one for E. coli? And would this help explain results of Fig. 3C?

P7L236. What about including just the growth of the WT cell (without the integrated cassette)? Why some growth curves are represented in Log scale (Fig. S4) and others not (Fig. 2d and 2e, Fig. S2, S3, S6)?

P7L242. Please, re-phrase that sentence!

P8L268. I miss here some type of conclusion for this part! It could be something that "the J105 configuration seems the best one and is the one we would use in the future for the dCas versions..." or similar! But conclude with something!! The same applies for P10L343!

Supplementary Fig. 10 Is not mentioned in the MS! In this Fig. the J105 strain without ara have a basal expression similar to that of the constitutive promoter PprsM. Any comment?

P9L295 (and Fig. 4). How does the different plant TF constructions affect cellular growth?

P10L330. So, the explanation proposed of why the JUB1 4x is worst that 2x is the increased spacing distance to allow activation by SoxS? That could be easily tested by introducing 2x and a buffer seq to simulate a 4x situation and see what is the output. Then, why the 4x is similar to 1x? As a curiosity, why authors did not test 3x?

P10L332. We know that constitute promoters expressing a reporter impose a high metabolic burden (specially in high copy number plasmids) and are far from ideal. What about a fairer comparison using a common expression system such as lacIq-Ptrc?

Related to that, what is the ori of the plasmids used? That is an important fact to take into account!

And, why in the DaraBAD, ParaE strain the constitutive promoter behaves as OK as the 1x?

Why MFI is represented in Fig.4C and a normalized one in 4d? Wouldn't it be clearer to show the data as in 4d?

P10L343. It would be nice to see a direct comparison of the best configuration of the dCas and the plant one.

P11L358. Typo! Probably authors mean to say "d". Similar typo in line 364. Also, the "D" symbol is missing in the chart 4d.

P11L362. M9 with what C-source?

P12L384. What variant of RamR did authors used? WT or one of the evolved variants described in <https://doi.org/10.1038/s41589-022-01072-w>. Please, specify!

P12L374. Maybe this is a naïve question but why do we want a Salmonella biosensor strain? Would not be better to use a non-pathogenic E. coli? Please, explain!

P12L393. What ara concentration was used to test the SALSOR 0.1 strain?

P12L397. I would like to have a clear explanation of the chart of Fig. 5b.

P13L424. Typo in the % of arabinose.

Fig. 5. Is the RFU/OD also normalized to the value obtained in the WT as indicated on the y-axis of the

chart? If so, explain that in the legend. In data S4 appears Fig.4 and not 5. I would be nice to see the output of just using the solvent DMSO.

Why authors did not test other (lower) concentrations of NOS, ROTU and TPH to test the proper functioning of the biosensor? With the experiments presented in Fig. 5 do not see how this biosensor would be able to detect low concentration of BIAs

P14L448. How stable is the strain with 5 copies of the very same sequence (JUB1 binding site, 40 bp)?

P14L459. And what would authors do to increase that?

Fig. 6. Why there is no difference between a plasmid without promoter (pKS2151200) and the one with a promoter (pKS2151201)?

RESPONSE LETTER

Referee expertise:

Referee #1: Synthetic biology

Referee #2: Biotechnology, synthetic biology and expression systems in gram-negative bacteria

Reviewers' comments:

Reviewer #1 (Remarks to the Author):

The revised manuscript provided by the authors represents a significant improvement over the original version, containing additional figures and explanations that address most of this reviewer's concerns. Notably, the authors have improved the maximum induction ratio of their aTF to 35-fold, sufficiently demonstrated tunable expression using their aTFs, and improved the performance of the alkaloid biosensors and carotenoid production strains. The text edits made to this revision help to clarify the novelty of the plant-based aTFs, better support the claims, and improve the accuracy of the manuscript as a whole. Overall, this work should be of interest to the microbial synthetic biology community.

The following minor edits are suggested to optionally improve the clarity of the manuscript.

1. Using the more common abbreviation THP for TetraHydro Papaverine, instead of TPH used throughout this manuscript, may improve clarity.

RESPONSE:

Thank you for this suggestion. We have changed the used abbreviation throughout the manuscript (page 13, lines 404 and 421; page 14, line 450, page 17, line 540; and page 20, line 631).

2. In the legend of figure 4, subpanel "e" is referenced, but does not exist. Should this instead be subpanel "d"?

RESPONSE:

The subpanel labeling has been corrected in **Fig. 4 (e to d, page 12, lines 379 and 385)**.

3. The language in the discussion (page 17, lines 543-555) should be toned down where appropriate. Statements such as "The generated ATFs can be adapted to different induction systems, including chemical inducers (e.g., tetracycline, IPTG) as well as optogenetic inducers" should instead indicate that the aTFs may prove possible to adapt, since the authors did not directly test the aTFs with these inducers.

RESPONSE:

As suggested, we have re-phrased the relevant sections in the "Discussion" (page 16, line 550, lines 552 - 556, 565 - 567).

Reviewer #3 (Remarks to the Author):

This MS explains the construction of several arabinose inducible expression systems based on dCas9 and plant-derived TF. Then, they validate those systems with two examples: i) building a biosensor strain and ii) constructing a strain to produce b-carotene.

In general terms the paper and the developed tools are interesting (even though they would not be my first choice) but it would be great to have a revised version of the MS addressing the following points: detailed and focused content and clearer conclusions.

RESPONSE:

We greatly appreciate the reviewer's comments and have taken them into consideration by performing several additional experiments to thoroughly characterize the developed arabinose regulatory toolkit and its viability and stability. As a result, we have incorporated new **Fig. 4e** and supplementary figures (**S11, S18, and S19**).

Below there are some comments and questions that need to be addressed before publication:

I would change the term "Gram-negative" in the title since this technology is only appropriate for just a limited number of gram-negatives, Enterobacteria.

RESPONSE:

Regarding the term "Gram-negative" in the title, to keep the title general, we did not change the title. However, in order to address the reviewer's comment, we now discuss the potential limitations of the developed ATFs as a versatile regulation toolkit for Gram-negative bacteria and specifically Enterobacteriaceae in the "Discussion" (page 18, lines 562 - 567).

The intro section should be shortened by going directly to the point! The necessity of expression systems to understand GRNs or to create biotech applications or for metabolic engineering.

RESPONSE:

In response to this comment, we have shortened the "Introduction" section (see page 3, lines 57-107).

What about the Anderson constitutive promoter library (doi:10.1186/1754-1611-3-4)?

RESPONSE:

Thanks for the comment! We have added the promoter library of Anderson *et al.* (2010, ref. 11, page 3, lines 64 – 66), who reported various synthetic elements, including (tet) promoters in *E. coli*.

What about commonly used expression system? Why are not mentioned in the MS?

RESPONSE:

Thanks for addressing this point. In addition to chemical inducers (such as tetracycline- and IPTG-inducible promoters), the (inducible) T7 RNA polymerase-based system is a common bacterial expression system. The T7 RNA polymerase-based expression system is widely used, as an orthogonal regulator for gene expression, in bacteria that typically is placed under the control of an inducible promoter, such as IPTG-inducible promoters. The generated ATFs in our study offer promising alternatives to adapt to the chemical inducer, as well as optogenetic inducer. By placing a single inducible promoter upstream of the ATFs, we should be able to express multiple proteins at different levels, which is crucial for bioengineering projects aimed at engineering lengthy biosynthetic pathways or studying complex gene regulatory networks. However, to successfully implement a new inducer to activate ATF expression, the chemical induction system requires to be adequately characterized. Furthermore, the ATF transcriptional output might be further fine-tuned using different gRNAs (for CRISPR/dCas9) or by altering the copy numbers of plant-derived TF binding sites. We should also highlight that a potential drawback is that the T7 promoter and T7 RNA polymerase system are specific to the T7 phage and may not be suitable for expressing all genes or proteins. Some genes may require additional modifications, such as the inclusion of specific regulatory elements, to achieve optimal expression levels or proper folding of the target protein that was highlighted in the revised manuscript.

In response to this comment of reviewer #3 and the editorial team, we compared our ATFs to the T7 expression system and other inducible expression systems in the “Discussion” (e.g. page 18, lines 554 - 562).

How stable are these constructs? It seems easy (many regulators involved) to get spontaneous mutant that STOP the entire expression system. Would that explain the low percentage of cells expressing the reporter?

RESPONSE:

Although the ATFs are relatively large, the core regulatory elements of the ATF system are the synthetic promoter harboring ATF binding sites that are short sequences (one copy of the binding site is 20 bp – 40 bp). Besides, to minimize the metabolic burden, we also used i) low-copy number plasmid for all reporter constructs (encoding synthetic promoter upstream to control the expression of the gene of interest) that harbored the origin of replication (ori) 29807 (derived from ori p15A, Strauch *et al.*, 2000) and ii) an arabinose induction system to separate growth and production phases. In response to this concern of the editorial team and reviewer #3, we added **Fig. 6e** (page 16, lines 487 -516), **Supplementary Fig. 11 (a and b)**, page 32, lines 976 - 999), and **Supplementary Fig.19 (a and b)**, page 37, lines 1086 - 1101).

We demonstrated stable protein expression levels under the control of our inducible synthetic transcription factors (ATFs) for at least 12h post-induction. Additionally, to ensure the stability of the microbial subculture, we measured the expression output of the strong ATFs in subsequent subcultures, and we did not observe a significant difference in its induction level after subcultures. We believe these additional experiments and analyses provide strong evidence to address the concerns raised by reviewer #3 and the editorial team. These data not only validate the stability of our system over time but also reinforce the reliability and scalability of our approach for potential biotechnological applications.

P3L102. If we want to use this system, do we have to engineer the L-ara pathway? And what exactly mean that?

RESPONSE:

To use the arabinose-inducible ATFs, there is no need to engineer the arabinose pathway. As shown in **Fig. 4c**, the wild-type *Salmonella* background that was co-transformed with the expression plasmid for JUB1-derived ATF and reporter plasmid harboring two copies of JUB1 binding site (JUB1-2x) displayed an inducible range of more than 29-fold when induced with 0.05% arabinose in M9 minimal medium (page 11, lines 333 – 337). When a very high expression level of a protein is desirable, an engineered arabinose pathway will be beneficial (resulting in 35-fold induction compared to non-induced conditions, **Fig. 4d** on pages 12 – 13, lines 364 – 394).

Would this ara system would work in other Gram-negative bacteria?

RESPONSE:

The primary goal of our study is to establish a versatile design principle for bacterial gene activation utilizing various heterologous DNA binding domains (DBDs) (e.g., DBD of TFs from plants that are evolutionarily distant from bacteria) and their specific binding sites. Therefore, they are orthogonal to not only *E. coli* and *Salmonella*, but also potentially other bacteria. Moreover, the SoxS activation domain (SoxS AD) was previously shown to work in Gram-positive bacteria *Paenibacillus polymyxa* (Schilling *et al.*, 2020). We therefore believe that our inducible ATFs

enable the development of controllable transcription regulatory systems in a wide range of bacteria, which are highly sought after by the bacterial synthetic biology community.

P7L229. Does this dCas9-SoxS *Salmonella* codon optimized results in a less optimal one for *E. coli*? And would this help explain results of Fig. 3C?

RESPONSE:

As reviewer #3 mentioned, we used the dCas9-SoxS *Salmonella* codon optimized to activate gene expression in *E. coli*, which might result in a less transcriptional output for CRISPR/dCas9-derived ATF in *E. coli* compared to *Salmonella*. This possibility is now mentioned on pages 7 - 8, lines 249 – 251 in the revised manuscript.

P7L236. What about including just the growth of the WT cell (without the integrated cassette)?

Why some growth curves are represented in Log scale (Fig. S4) and others not (Fig. 2d and 2e, Fig. S2, S3, S6)?

RESPONSE:

We included the growth curve of WT cells in **Supplementary Fig. 4** (pages 27 – 28, lines 890 - 901). Moreover, we now show all growth curves (**Fig. 2d, Fig. 2e, Supplementary Fig. 2, Supplementary Fig. 3, Supplementary Fig. 4, Supplementary Fig. 6, and Supplementary Fig. 18**) using OD₆₀₀ scale data (not Log OD₆₀₀).

P7L242. Please, re-phrase that sentence!

RESPONSE:

The sentence was re-phrased in the revised manuscript (page 7, lines 236 - 279).

P8L268. I miss here some type of conclusion for this part! It could be something that "the J105 configuration seems the best one and is the one we would use in the future for the dCas versions..." or similar! But conclude with something!! The same applies for P10L343!

RESPONSE:

In response to this comment, we added a conclusion to the end of sections "**Arabinose-inducible CRISPR/dCas9-derived ATF library**" (page 8, lines 263 - 277) and "**Arabinose-inducible plant-derived ATF library**" (page 11, lines 356 - 363).

Supplementary Fig. 10 Is not mentioned in the MS! In this Fig. the J105 strain without ara have a basal expression similar to that of the constitutive promoter P_{prsm}. Any comment?

RESPONSE:

Thank you for pointing this out. The figure is now mentioned in the manuscript (**Supplementary Fig. 7** of the revised manuscript, page 7, line 245).

The strength of the ATF can influence basal expression. We speculate that we observe a high basal expression, similar to constitutive *P_{prsm}* promoter, because of the high efficiency of CRISPR0/dCas9-derived ATF targeting J105 also at very low dCas9 expression levels. In fact, due to the possibly leaky expression from the arabinose-inducible promoter, some dCas9 proteins are present in the cell also in the absence of the inducer. Since the gRNA targeting *J105* (within the *J1* synthetic promoter located upstream of *mRFP1*) is constitutively expressed, as a result, some transcription might occur also in the absence of dCas9 inducer.

P9L295 (and Fig. 4). How does the different plant TF constructions affect cellular growth?

RESPONSE:

We added new **Supplementary Fig. 18** (pages 36 – 37, lines 1072 -1084) to address the concern of reviewer #3 on the time-course performance of plant-derived ATFs. Our results showed that JUB1-derived ATFs resulted in a minimal growth defect in the inducing mediums compared to the non-inducing medium. The differences in the time-course performance of JUB1-1x, JUB1-2x, and JUB1-4x might be due to the basal expression (see the expression level in the non-inducing medium shown in **Fig. 4c and d**, pages 12 – 13, lines 364 - 394)

P10L330. So, the explanation proposed of why the JUB1 4x is worst that 2x is the increased spacing distance to allow activation by SoxS? That could be easily tested by introducing 2x and a buffer seq to simulate a 4x situation and see what is the output. Then, why the 4x is similar to 1x?

RESPONSE:

According to Dong *et al.* (2022), a narrow region from -80 to -90 upstream of the transcription start site on the non-sense strand and from -50 to -80 on the sense strand results in effective CRISPR activation in *E. coli* (page 7, lines 239 - 242). Within the synthetic promoter of plant-derived ATF, 80 bp upstream of the 2x binding site in the reporter plasmid can be considered a buffer sequence.

As reviewer #3 mentioned, the fluorescent output of ATF targeting 4x binding sites is similar to that of ATF targeting 1x binding sites in *Salmonella*. In contrast, in *E. coli*, the fluorescent output of ATF targeting 4x binding sites is higher than that of ATF targeting 1x binding sites. Our data showed that increasing a synthetic transcription factor's

binding site copy number does not correlate linearly with its transcriptional output. The possible explanations might be:

1. Transcriptional regulation often involves cooperative interactions between transcription factors and their binding sites. Binding one transcription factor to a binding site may enhance the binding affinity or activity of another nearby transcription factor (as we observe for two copies of the binding site). However, increasing the copy number of binding sites from two to four may suppress the transcriptional output of the system.
2. Transcriptional regulation is a dynamic process involving multiple steps, including binding transcription factors to DNA, recruitment of transcriptional machinery, and transcription initiation. Increasing the binding site copy number may not linearly affect transcriptional output if other steps in the transcriptional process become rate-limiting. For example, if the recruitment of RNA polymerase is slower or limited, increasing the binding site copy number may not significantly enhance transcriptional output beyond a certain threshold.

As a curiosity, why authors did not test 3x?

RESPONSE:

We previously constructed expression plasmids for plant-derived ATFs and reporter plasmids for 1x, 2x, and 4x binding sites to activate gene expression in yeasts (Naseri *et al.*, 2021). We used these reporter plasmids to generate the reporter constructs in this study (**Supplementary Methods**, page 6, lines 212 – 218). Therefore, we did not develop reporter plasmids for the 3x binding site in this study.

P10L332. We know that constitute promoters expressing a reporter impose a high metabolic burden (specially in high copy number plasmids) and are far from ideal. What about a fairer comparison using a common expression system such as lacIq-Ptrc? Related to that, what is the ori of the plasmids used? That is an important fact to take into account! And, why in the DaraBAD, ParaE strain the constitutive promoter behaves as OK as the 1x?

RESPONSE:

In *Salmonella*, one of the commonly used promoters is the 5' region of *rpsM* (P_{rpsM}) (Cooper *et al.*, 2017). Therefore, we decided to use this promoter as a control. To avoid the metabolic burden issue, we used low-copy number plasmid for all reporter plasmids that harbor the origin of replication (ori) 29807 (derived from ori p15A, Strauch *et al.*, 2000). In response to this comment, we added the information regarding the plasmid copy number to the revised manuscript (page 10, line 318; page 10, line 319) and "Supplementary Methods" (page 3, lines 101 - 102; page 6, lines 205 – 207).

Regarding the transcriptional output of constitutive promoter compared to JUB-derived ATF targeting synthetic promoter with one copy of its binding site (1x), we obtained similar transcriptional output in the optimized background ($\Delta araBAD$, $P_{araE}::P_{araA}$) (**Fig. 4**, pages 12 - 13, lines 364 - 394). According to our data shown in **Fig. 2** (pages 6 - 7, lines 188 – 213), more sfGFP is expressed in the optimized background compared to the wild-type. Therefore, in the optimized background, we expect to see a higher level of JUB1-derived ATF expression compared to the wild-type background. However, the strain only has one copy of the JUB1 binding site upstream of the fluorescent protein. Because of this, it is possible that not all JUB1-derived ATFs will be captured by the synthetic promoter. In fact, there are more JUB1-derived ATF molecules inside the cell than what is actually needed to fully saturate the synthetic promoter containing just one copy of its binding site. Therefore, we speculate that JUB-1x could not activate reporter gene expression more than P_{rpsM} . However, further investigation would be needed to examine the system's behavior upon introducing ATFs, which is beyond the scope of the submitted manuscript.

Why MFI is represented in Fig.4C and a normalized one in 4d? Wouldn't it be clearer to show the data as in 4d?

RESPONSE:

In **Fig. 4c**, we used FACS flow cytometry to quantify ATF fluorescent output in wild-type *Salmonella*. This allows us to measure the mean fluorescent intensity of precisely the same number of cells for all samples (10,000 cells). However, to further prove the system output, we used a plate reader to measure the whole population fluorescent intensity and growth for the same ATFs of **Fig. 4c**, that was shown in **Fig. 4d** (wild-type that harbors native *araBAD* and P_{araE}). In addition, we included the plate reader data for those ATFs tested in the modified *Salmonella* background, bearing $\Delta araBAD$ and P_{proA} upstream of the *araE* gene (not native P_{araE}). To avoid confusion, in a new **Fig 4c** and **d**, we used "wild-type" instead of "*araBAD*, P_{araE} ".

P10L343. It would be nice to see a direct comparison of the best configuration of the dCas and the plant one.

RESPONSE:

We tested the ATFs in various backgrounds, *E. coli* and *Salmonella*, wild-type, and engineered for arabinose catabolism. Therefore, we were not able to show them all in a single figure. However, in response to this comment and another comment from reviewer #3 on the stability of the generated strains, we included new **Supplementary Fig. 11** (page 32, lines 976 - 999) and **Supplementary Fig. 19** (page 37, lines 1086 - 1101) that allows seeing a direct comparison of strong CRISPR/dCas9-derived ATF targeting J105 and JUB1-derived ATF targeting 2x binding site.

Supplementary Fig. 11

Supplementary Fig. 19

P11L358. Typo! Probably authors mean to say "d". Similar typo in line 364. Also, the "D" symbol is missing in the chart 4d.

RESPONSE:

Thank you for pointing this out. Corrected on pages 12 - 13, lines 376 - 392.

P11L362. M9 with what C-source?

RESPONSE:

The M9 minimal medium was supplemented with 0.4% glycerol and 0.1% casamino acids. We provided a detailed description of the M9 minimal medium in the "Method" section, page 19, lines 600 - 603. In response to this question, we now added the C-source used for M9 minimal medium to the manuscript as follows: page 9, lines 294 - 295; page 13, line 389; page 14, lines 448 - 449; page 26, lines 868 - 869; page 29, line 922; page 31, line 961; page 32, line 987; page 34, lines 1011-1012; page 35, lines 1050 - 1051, page 37, line 1080; and page 37, line 1096.

P12L384. What variant of RamR did authors used? WT or one of the evolved variants described in

<https://doi.org/10.1038/s41589-022-01072-w>. Please, specify!

RESPONSE:

Wide-type RamR gene was used in this study. In response to this comment, we changed the description of RamR in the revised manuscript (page 13, line 407).

P12L374. Maybe this is a naïve question but why do we want a *Salmonella* biosensor strain? Would not be better to use a non-pathogenic *E. coli*? Please, explain!

RESPONSE:

Additionally, depending on the specific needs and goals of the biosensor application, either *Salmonella* or non-pathogenic *E. coli* can be chosen as the biosensor strain.

E. coli continues to play a pivotal role in advancing the field of metabolic engineering and biotechnology due to its ease of manipulation, rapid growth, and well-established genetics. However, *E. coli* has a low tolerance towards high metabolic fluxes. Division of labor between bacterial strains of synthetic microbial consortia (SMC), *i.e.*, heterologous metabolite production by one bacteria and biosensor expression by the other bacteria, reduces the metabolic burden, improving the system's overall productivity. As discussed on page 18, lines 543 - 547, an interesting potential application of SALSOR 0.2 strain will be its application as the whole-cell biosensor for quantification of the metabolite, for example, produced by neighbor *E. coli* cells in SMC.

Salmonella has evolved to detect and respond to external signals in complex environments, such as the host during infection. This natural sensing ability can be harnessed and genetically engineered to create biosensors that respond to specific target molecules or environmental conditions. Establishing (potentially attenuated) *Salmonella* strains as a sensitive biosensor using our ATFs might have applications in environmental monitoring, food safety, and medical diagnostics.

P12L393. What ara concentration was used to test the SALSOR 0.1 strain?

RESPONSE:

We used the LB medium without arabinose inducer in the case of SALSOR 0.1, as the constitutive promoter P_{RamR} was used to express *RamR*. In the case of SALSOR 0.2, 0.05% arabinose was used to induce the expression of plant-derived ATF. To clarify this, the information was added to the legend of **Fig. 5** (page 14, lines 447 and 449).

P12L397. I would like to have a clear explanation of the chart of Fig. 5b.

RESPONSE:

A proper explanation was added to the manuscript, as it was suggested by reviewer #3 (page 13, lines 410-414).

P13L424. Typo in the % of arabinose.

RESPONSE:

Corrected on page 14, line 448.

Fig. 5. Is the RFU/OD also normalized to the value obtained in the WT as indicated on the y-axis of the chart? If so, explain that in the legend. In data S4 appears Fig.4 and not 5. I would be nice to see the output of just using the solvent DMSO.

RESPONSE:

RFU data was normalized to the value obtained in the WT, which was explained in the legend of **Fig. 5** in the revised manuscript. The title of source **Data S4** was corrected. Additionally, the data of the WT sample was added to **Data S4**. We used WT (without biosensor cassette) to normalize the data. To prepare the WT sample, we used DMSO without ligands, added to the legend of **Fig. 5** in the revised manuscript (page 14, lines 451 – 452).

Why authors did not test other (lower) concentrations of NOS, ROTU and TPH to test the proper functioning of the biosensor? With the experiments presented in Fig. 5 do not see how this biosensor would be able to detect low concentration of BIAs

RESPONSE:

We agree that testing various concentrations of ligands will be interesting. However, this was not the research goal of the present study. Here, we aimed to establish a collection of versatile ATFs that can be used for various synthetic biology applications in bacteria. As a proof of concept for applying plant-derived ATFs in *Salmonella*, we established a sensitive biosensor compared to the available biosensor, reported by d'Oelsnitz *et al.* (2022). We intend to test various concentrations of the ligands as reported by d'Oelsnitz *et al.* (2022) in future work.

P14L448. How stable is the strain with 5 copies of the very same sequence (JUB1 binding site, 40 bp)?

RESPONSE:

Using inducible promoters, we postpone the expression of JUB1-derived ATF to the production phase to minimize the unwanted metabolic burden on the cell. This might support the stability of the strain. To address this concern, we measured the production of β -carotene in the clone after four subsequent rounds to assess the system's long-term stability in the presence of 0.2% arabinose (**Fig. 6e**, page 16, lines 504 – 507). This data shows no significant difference between different sub-cultures, confirming that the short repeat sequence DNA sequence inside the cell does not influence the strain's genetic stability (see also page 15, line 481).

P14L459. And what would authors do to increase that?

RESPONSE:

We modified the sentence and replaced the "increased" with "high".

Page 15, lines 482 - 485: "These data suggest a high capacity of the arabinose-inducible JUB1-derived ATF to increase the transcription level of the crtEBIY operon and β -carotene production compared to expression from the constitutive promoter (**Fig. 6d**)".

Fig. 6. Why there is no difference between a plasmid without promoter (pKS2151200) and the one with a promoter (pKS2151201)?

RESPONSE:

This is an excellent question and it would be interesting to investigate this observation in further detail in future work. It appears to be possible that also pKS2151200 harbors a weak promoter upstream of the β -carotene biosynthetic pathway genes, which is not annotated in the plasmid. Both pKS2151200 and pKS2151201 plasmids were constructed by the Glasgow iGEM 2016 team and we did not re-sequence the plasmids. Importantly, however, using our JUB1-derived ATF, we do observe a more than 2-fold increase in expression, validating that plant-derived ATFs can be used for high-efficiency production of metabolites in *E. coli*.

REVIEWERS' COMMENTS:

Reviewer #3 (Remarks to the Author):

The authors have responded comprehensively to all the questions I raised, resulting in a substantial improvement in the manuscript. However, I maintain my suggestion that the title should emphasize Enterobacteria rather than being too broadly focused on Gram-negative bacteria. If this is not possible, I would recommend that the abstract and discussion clearly highlight that the "primary objective of the study is to establish a versatile design principle for bacterial gene activation," as the authors have stated in their rebuttal letter.

Minor comment, in the x-axis of Fig. 6e appears 8h but in the legend 4h is mentioned. Please fix the typo.

Nonetheless, I would like to extend my congratulations to the authors for their work on this manuscript.

RESPONSE LETTER

REVIEWERS' COMMENTS:

Reviewer #3 (Remarks to the Author):

The authors have responded comprehensively to all the questions I raised, resulting in a substantial improvement in the manuscript. However, I maintain my suggestion that the title should emphasize Enterobacteria rather than being too broadly focused on Gram-negative bacteria. If this is not possible, I would recommend that the abstract and discussion clearly highlight that the "primary objective of the study is to establish a versatile design principle for bacterial gene activation," as the authors have stated in their rebuttal letter.

RESPONSE:

As the reviewer suggested, we now rephrased the title to include "Enterobacteriaceae" in the revised manuscript. We also replaced "Gram-negative bacteria" with "Enterobacteriaceae" throughout the manuscript, if needed. For details, see

page 1, page 2, page 3, page 11, page 20, and page 21.

Minor comment, in the x-axis of Fig. 6e appears 8h but in the legend 4h is mentioned. Please fix the typo.

RESPONSE:

Sorry for the confusion! We measured the carotenoid production at time points 8h, 12h, 24h, and 36h. We corrected the legend of **Fig. 6e** in the revised manuscript (page 23).

Nonetheless, I would like to extend my congratulations to the authors for their work on this manuscript.

RESPONSE:

We appreciate your positive feedback.